# Neural Networks with Small Weights and Depth-Separation Barriers

**Gal Vardi**
Weizmann Institute of Science
gal.vardi@weizmann.ac.il

**Ohad Shamir**
Weizmann Institute of Science
ohad.shamir@weizmann.ac.il

## Abstract

In studying the expressiveness of neural networks, an important question is whether there are functions which can only be approximated by sufficiently deep networks, assuming their size is bounded. However, for constant depths, existing results are limited to depths 2 and 3, and achieving results for higher depths has been an important open question. In this paper, we focus on feedforward ReLU networks, and prove fundamental barriers to proving such results beyond depth $4$, by reduction to open problems and natural-proof barriers in circuit complexity. To show this, we study a seemingly unrelated problem of independent interest: Namely, whether there are polynomially-bounded functions which require super-polynomial weights in order to approximate with constant-depth neural networks. We provide a negative and constructive answer to that question, by showing that if a function can be approximated by a polynomially-sized, constant depth $k$ network with arbitrarily large weights, it can also be approximated by a polynomially-sized, depth $3k + 3$ network, whose weights are polynomially bounded.

## 1   Introduction

The *expressive power* of feedforward neural networks has been extensively studied in recent years. It is well-known that sufficiently large depth-2 neural networks, using reasonable activation functions, can approximate any continuous function on a bounded domain ([5, 9, 15, 2]). However, the required size of such networks can be exponential in the input dimension, which renders them impractical. From a learning perspective, both theoretically and in practice, the main interest is in neural networks whose size is at most polynomial in the input dimension.

When considering the expressive power of neural networks of bounded size, a key question is what are the tradeoffs between the width and the depth. Overwhelming empirical evidence indicates that deeper networks tend to perform better than shallow ones, a phenomenon supported by the intuition that depth, providing compositional expressibility, is necessary for efficiently representing some functions. From the theoretical viewpoint, quite a few works in the past few years have explored the beneficial effect of depth on increasing the expressiveness of neural networks. A main focus is on *depth separation*, namely, showing that there is a function $f : \mathbb{R}^d \to \mathbb{R}$ that can be approximated by a $\text{poly}(d)$-sized network of a given depth, with respect to some input distribution, but cannot be approximated by $\text{poly}(d)$-sized networks of a smaller depth. Depth separation between depth 2 and 3 was shown by [7] and [6]. However, despite much effort, no such separation result is known for any constant greater than 2. Thus, it is an open problem whether there is separation between depth 3 and some constant depth greater than 3. Separation between networks of a constant depth and networks with $\text{poly}(d)$ depth was shown by [30] (see related work section below for more details).

In fact, a similar question has been extensively studied by the theoretical computer science community over the past decades, in the context of Boolean and threshold circuits of bounded size. Showing limitations for the expressiveness of such circuits (i.e. *circuit lower bounds*) can contribute to our

understanding of the $P \neq NP$ question, and can have other significant theoretical implications [1]. Despite many attempts, the results on circuit lower bounds were limited. In a seminal work, [23] described a main technical limitation of current approaches for proving circuit lower bounds: They defined a notion of "natural proofs" for a circuit lower bound (which include current proof techniques), and showed that obtaining lower bounds with such proof techniques would violate a widely accepted conjecture, namely, that pseudorandom functions exist. This *natural-proof barrier* explains the lack of progress on circuit lower bounds. More formally, they show that if a class $\mathcal{C}$ of circuits contains a family of pseudorandom functions, then showing for some function $f$ that $f \notin \mathcal{C}$ cannot be done with a natural proof. As a result, if we consider the class $\mathcal{C}$ of $\text{poly}(d)$-sized circuits of some bounded depth $k$, where $k$ is large enough so that $\mathcal{C}$ contains a pseudorandom function family, then it will be difficult to show that some functions are not in $\mathcal{C}$, and hence that these functions require depth larger than $k$ to express.

An object closer to actual neural networks are threshold circuits. These are essentially neural networks with a threshold activation function in all neurons (including the output neuron), and where the inputs are in $\{0, 1\}^d$. The problem of depth separation in threshold circuits was widely studied [22]. This problem requires, for some integer $k$, a function that cannot be computed by a threshold circuit of width $\text{poly}(d)$ and depth $k$, but can be computed[1] by a threshold circuit of width $\text{poly}(d)$ and depth $k' > k$. [20] and [16] showed a candidate pseudorandom function family computable by threshold circuits of depth 4, width $\text{poly}(d)$, and $\text{poly}(d)$-bounded weights. By [23], it implies that for every $k' > k \geq 4$, there is a natural-proof barrier for showing depth separation between threshold circuits of depth $k$ and depth $k'$. As for smaller depths, a separation between threshold circuits of depth 3 and some $k > 3$ is a longstanding open problem (although there is no known natural-proof barrier in this case), and separation between threshold circuits of depth 2 and 3 is known under the assumption that the weight magnitudes are $\text{poly}(d)$ bounded [14].

Since a threshold circuit is a special case of a neural network with threshold activation and where the inputs and output are Boolean, it is natural to ask whether the barriers to depth separation in threshold circuits have implications on the problem of depth separation in neural networks. Such implications are not obvious, since neural networks have real-valued inputs and outputs (not necessarily just Boolean ones), and a continuous activation function. Thus, it might be possible to come up with a depth-separation result, which crucially utilizes some function and inputs in Euclidean space. In fact, this can already be seen in existing results: For example, separation between threshold circuits of constant depth ($\text{TC}^0$) and threshold circuits of $\text{poly}(d)$ depth (which equals the complexity class P/poly) is not known, but [30] showed such a result for neural networks. His construction is based on the observation that for one dimensional data, a network of depth $k$ is able to express a sawtooth function on the interval $[0, 1]$ which oscillates $\mathcal{O}(2^k)$ times. Clearly, this utilizes the continuous structure of the domain, in a way that is not possible with Boolean inputs. Also, the depth-2 vs. 3 separation results of [7] and [6] rely on harmonic analysis of real functions. Finally, the result of [7] does not make any assumption on the weight magnitudes, whereas relaxing this assumption for the parallel result on threshold circuits is a longstanding open problem [22].

**Main Result 1: Barriers to Depth Separation**

In this work, we focus on real-valued neural networks with the ReLU activation function, and show (under some mild assumptions on the input distribution and on the function) that any depth-separation result between neural networks of depth $k \geq 4$ and some constant $k' > k$, would imply depth separation between threshold circuits of depth $k - 2$ and some constant greater than $k - 2$. Hence, showing depth separation with $k = 5$ would solve the longstanding open problem of separating between threshold circuits of depth 3 and some larger constant. Showing depth separation with $k \geq 6$ would solve the open problem of separating between threshold circuits of depth $k - 2$ and some larger constant, which is especially challenging due to the natural-proof barrier for threshold circuits of depth at least 4. Finally, showing depth separation with $k = 4$ would solve the longstanding open problem of separating between threshold circuits of depth 2 (with arbitrarily large weights) and some larger constant (we note that separation between threshold circuits of depth 2 and 3 is known only under the assumption that the weight magnitudes are $\text{poly}(d)$ bounded). The result applies to both

continuous and discrete input distributions. Thus, we show a barrier to depth separation, that explains the lack of progress on depth separation for constant-depth neural networks of depth at least $4$.

While this is a strong barrier to depth separation in neural networks, it should not discourage researchers from continuing to investigate it. First, our results focus on plain feedforward ReLU networks, and do not necessarily apply to other architectures. Second, we do make assumptions on the input distribution and the function, which are mild but perhaps can be circumvented (or alternatively, relaxed). Third, our barrier does not apply to separation between depth $3$ and some larger constant. That being said, we do show that in order to achieve separation between depth $k \geq 3$ and some constant $k' > k$, some different approach than these used in current results would be required. As far as we know, in all existing depth-separation results for continuous input distributions (e.g., [7, 6, 30, 24, 17, 32, 25]) the functions are either of the form $f(\mathbf{x}) = g(\|\mathbf{x}\|)$ or $f(\mathbf{x}) = g(x_1)$ for $g : \mathbb{R} \to \mathbb{R}$. Namely, $f$ is either radial or depends on one component[2]. We show that for such functions, networks of a constant depth greater than $3$ do not have more power than depth-$3$ networks.

**Main Result 2: Effect of Weight Magnitude on Expressiveness**

To establish our depth-separation results, we actually go through a seemingly unrelated problem of independent interest: Namely, what is the impact on expressiveness if we force the network weights to have reasonably bounded weights (say, $\mathrm{poly}(d)$). This is a natural restriction: Exponentially-large weights are unwieldy, and moreover, most neural networks used in practice have small weights, due to several reasons related to the training process, such as regularization, standard initialization of the weights to small values, normalization heuristics, and techniques to avoid the exploding gradient problem [13]. Therefore, it is natural to ask how bounding the size of the weights affects the expressive power of neural networks. As far as we know, there are surprisingly few works on this, and current works on the expressiveness of neural networks often assume that the weights may be arbitrarily large, although this is not the case in practice.

If we allow arbitrary functions, there are trivial cases where limiting the weight magnitudes hurts expressiveness. For example, let $f : [0,1]^d \to \mathbb{R}$, where for every $\mathbf{x} = (x_1, \ldots, x_d)$ we have $f(\mathbf{x}) = x_1 \cdot 2^d$. Clearly, $f$ can be expressed by a network of depth $1$ with exponential (in $d$) weights. This function cannot be approximated w.r.t. the uniform distribution on $[0,1]^d$ by a constant-depth network with $\mathrm{poly}(d)$ width and $\mathrm{poly}(d)$-bounded weights, since such networks cannot compute exponentially-large values. However, functions of practical interest only have constant or $\mathrm{poly}(d)$-sized values (or at least can be well-approximated by such functions). Thus, a more interesting question is whether for approximating such functions, we may need weights larger than $\mathrm{poly}(d)$.

In our paper, we provide a negative answer to this question, in the following sense: Under some mild assumptions on the input distribution, if the function can be approximated by a network with ReLU activation, width $\mathrm{poly}(d)$, constant depth $k$ and arbitrarily large weights, then we show how it can be approximated by a network with ReLU activation, width $\mathrm{poly}(d)$, depth $3k+3$, and weights bounded by $\mathrm{poly}(d)$ or by a constant. The result applies to both continuous and discrete input distributions.

The two problems that we consider, namely depth-separation and the power of small weights, may seem unrelated. Indeed, each problem considers a different aspect of expressiveness in neural networks. However, perhaps surprisingly, the proofs for our results on barriers to depth separation follow from our construction of networks with small weights. In a nutshell, the idea is that our deeper small-weight network is such that most layers implement a threshold circuit. Thus, if we came up with a "hard" function $f$ that provably requires much depth to express with a neural network, then the threshold circuit used in expressing it (via our small-weight construction) also provably requires much depth – since otherwise, we could make our small-weight network shallower, violating the assumption on $f$. This would lead to threshold-circuit lower bounds.

**Related Work**

**Depth separation in neural networks.** As we already mentioned, depth separation between depth $2$ and $3$ was shown by [7] and [6]. In [7] there is no restriction on the weight magnitudes of the depth-$2$ network, while [6] assumes that the weights are bounded by $2^d$. The input distributions there are

continuous. A separation result between depth 2 and 3 for discrete inputs is implied by [19], for the function that computes inner-product mod 2 on binary vectors (see also a discussion in [7]).

In [30], it is shown that there exists a family of univariate functions $\{\varphi_k\}_{k=1}^{\infty}$ on the interval $[0,1]$, such that for every $k$ the function $\varphi_k$ can be expressed by a network of depth $k$ and width $\mathcal{O}(1)$, but cannot be approximated by any $o(k/\log(k))$-depth, $\mathrm{poly}(k)$-width network w.r.t. the uniform distribution on $[0,1]$. To rewrite this as a depth separation result in terms of a dimension $d$, let $\{f_d\}_{d=1}^{\infty}$ where $f_d : [0,1]^d \to \mathbb{R}$ is such that $f_d(\mathbf{x}) = \varphi_d(x_1)$. The result of [30] implies that $f_d$ can be expressed by a network of width $\mathcal{O}(1)$ and depth $d$, but cannot be approximated by a network of width $\mathrm{poly}(d)$ and constant depth. Hence, there is separation between constant and polynomial depths. However, it does not have implications for the problem of separating between constant depths.

In [24, 17, 32] another notion of depth separation is considered. They show that there are functions that can be $\epsilon$-approximated by a network of $\mathrm{polylog}(1/\epsilon)$ width and depth, but cannot be $\epsilon$-approximated by a network of $\mathcal{O}(1)$ depth unless its width is $\mathrm{poly}(1/\epsilon)$. Their results are based on a univariate construction similar to the one in [30].

**Expressive power of neural networks with small weights.** [18] considered a neural network $N$ with a piecewise linear activation function in all hidden neurons, and threshold activation in the output neuron. Namely, $N$ computes a Boolean function. He showed that if every hidden neuron in $N$ has fan-out 1, and the $d$-dimensional input is from a certain discrete set, then there is a network $N'$ of the same size and same activation functions, that computes the same function, and its weights and biases can be represented by $\mathrm{poly}(d)$ bits. Thus, the weights in $N'$ are bounded by $2^{\mathrm{poly}(d)}$. From his result, it is not hard to show the following corollary: Let $N$ be a network with ReLU activation in all hidden neurons and threshold activation in the output neuron, and assume that the input to $N$ is from $\{0,1\}^d$, and that $N$ has width $\mathrm{poly}(d)$ and constant depth. Then, there is a threshold circuit of $\mathrm{poly}(d)$ width, constant depth, and $\mathrm{poly}(d)$-bounded weights, that computes the same function. Note that this result considers exact computation of functions with binary inputs and output, while we consider approximation of functions with real inputs and output.

Expressiveness with small weights was also studied in the context of threshold circuits. In particular, it is known that every function computed by a polynomial-size threshold circuit of depth $k$ can be computed by a polynomial-size threshold circuit of depth $k+1$ with weights whose absolute values are bounded by a polynomial or a constant ([11, 10, 27]). This result relies on the fact that threshold circuits compute Boolean functions and does not apply to real-valued neural networks.

In the *weight normalization* method ([26]), the weights are kept normalized during the training of the network. That is, all weight vectors of neurons in the network have the same Euclidean norm. Some approximation properties of such networks were studied in [31]. The dependence of the sample complexity of neural networks on the norms of its weight matrices was studied in, e.g., [3, 12, 21].

Our paper is structured as follows: In Section 2 we provide notations and definitions, followed by our results in Section 3. We sketch our proof ideas in Section 4, with all proofs deferred to Appendix A.

## 2  Preliminaries

**Notations.** We use bold-faced letters to denote vectors, e.g., $\mathbf{x} = (x_1, \dots, x_d)$. For $\mathbf{x} \in \mathbb{R}^d$ we denote by $\|\mathbf{x}\|$ the Euclidean norm. For a function $f : \mathbb{R}^d \to \mathbb{R}$ and a distribution $\mathcal{D}$ on $\mathbb{R}^d$, either continuous or discrete, we denote by $\|f\|_{L_2(\mathcal{D})}$ the $L_2$ norm weighted by $\mathcal{D}$, namely $\|f\|_{L_2(\mathcal{D})}^2 = \mathbb{E}_{\mathbf{x} \sim \mathcal{D}}(f(\mathbf{x}))^2$. For a set $A$ we let $\mathbb{1}_A$ denote the indicator function. For an integer $d \geq 1$ we denote $[d] = \{1, \dots, d\}$.

**Neural networks.** We consider feedforward neural networks, computing functions from $\mathbb{R}^d$ to $\mathbb{R}$. The network is composed of layers of neurons, where each neuron computes a function of the form $\mathbf{x} \mapsto \sigma(\mathbf{w}^\top \mathbf{x} + b)$, where $\mathbf{w}$ is a weight vector, $b$ is a bias term and $\sigma : \mathbb{R} \mapsto \mathbb{R}$ is a non-linear activation function. In this work we focus on the ReLU activation function, namely, $\sigma(z) = [z]_+ = \max\{0, z\}$. For a matrix $W = (\mathbf{w}_1, \dots, \mathbf{w}_n)$, we let $\sigma(W^\top \mathbf{x} + \mathbf{b})$ be a shorthand for $(\sigma(\mathbf{w}_1^\top \mathbf{x} + b_1), \dots, \sigma(\mathbf{w}_n^\top \mathbf{x} + b_n))$, and define a layer of $n$ neurons as $\mathbf{x} \mapsto \sigma(W^\top \mathbf{x} + \mathbf{b})$. By denoting the output of the $i$-th layer as $O_i$, we can define a network of arbitrary depth recursively by $O_{i+1} = \sigma(W_{i+1}^\top O_i + \mathbf{b}_{i+1})$. The *weights vector* of the $j$-th neuron in the $i$-th layer is the $j$-th column of $W_i$, and its *outgoing-weights vector* is the $j$-th row of $W_{i+1}$. The *fan-in* (respectively, *fan-out*) of a neuron is the number of non-zero entries in its weights vector (respectively, outgoing-weights

vector). The final layer $h$ is purely linear with no bias, i.e. $O_h = W_h^\top \cdot O_{h-1}$. We define the *depth* of the network as the number of layers $l$, and denote the number of neurons $n_i$ in the $i$-th layer as the *size* of the layer. We define the *width* of a network as $\max_{i \in [l]} n_i$. We sometimes consider neural networks with multiple outputs. We say that a neural network has $\mathrm{poly}(d)$-*bounded weights* if for all individual weights $w$ and biases $b$, the absolute values $|w|$ and $|b|$ are bounded by some $\mathrm{poly}(d)$.

**Threshold circuits.** A threshold circuit is a neural network with the following restrictions: (1) The activation function in all neurons is $\sigma(z) = \mathrm{sign}(z)$. We define $\mathrm{sign}(z) = 0$ for $z \leq 0$, and $\mathrm{sign}(z) = 1$ for $z > 0$. A neuron in a threshold circuit is called a *threshold gate*. (2) The output gates also have a $\mathrm{sign}$ activation function. Hence, the output is binary. (3) We always assume that the input to a threshold circuit is a binary vector $\mathbf{x} \in \{0, 1\}^d$. (4) Since every threshold circuit with real weights can be expressed by a threshold circuit of the same size with integer weights (c.f. [10]), we assume w.l.o.g. that all weights are integers.

**Probability densities.** Let $\mu$ be the density function of a continuous distribution on $\mathbb{R}^d$. For $i \in [d]$ we denote by $\mu_i$ and $\mu_{[d]\setminus i}$ the marginal densities for $x_i$ and $\{x_1, \ldots, x_{i-1}, x_{i+1}, \ldots, x_d\}$ respectively. We denote by $\mu_{i|[d]\setminus i}$ the conditional density of $x_i$ given $\{x_1, \ldots, x_{i-1}, x_{i+1}, \ldots, x_d\}$. Thus, for every $i$ and $\mathbf{x}^i = (x_1, \ldots, x_{i-1}, x_{i+1}, \ldots, x_d)$ we have $\mu(\mathbf{x}) = \mu_{[d]\setminus i}(\mathbf{x}^i)\mu_{i|[d]\setminus i}(x_i|\mathbf{x}^i)$. We say that $\mu$ has an *almost-bounded support*, if for every $\delta = \frac{1}{\mathrm{poly}(d)}$ there is $R = \mathrm{poly}(d)$, such that $Pr_{\mathbf{x} \sim \mu}(\mathbf{x} \notin [-R, R]^d) \leq \delta$. We say that $\mu$ has an *almost-bounded conditional density* if for every $\epsilon = \frac{1}{\mathrm{poly}(d)}$ there is $M = \mathrm{poly}(d)$ such that for every $i \in [d]$ we have $Pr_{\mathbf{x} \sim \mu}\left(\sup_{t \in \mathbb{R}} \mu_{i|[d]\setminus i}(t|x_1, \ldots, x_{i-1}, x_{i+1}, \ldots, x_d) > M\right) \leq \epsilon$.

**Remark 2.1.** *In our results on continuous distributions we assume that the density $\mu$ has an almost-bounded support and an almost-bounded conditional density. While the first assumption is intuitive, the second is less standard. However, it is mild and intended to exclude distributions which are both continuous and with significant mass on extremely small domains. In Appendix B we show that it holds, for example, for Gaussians (as long as the variance is at least $1/\mathrm{poly}(d)$ in all directions), mixtures of Gaussians, any distribution after a Gaussian smoothing, the uniform distribution on a ball, as well as distributions from existing depth-separation results. In addition, with a slightly different proof, we also provide similar results for discrete distributions.*

**Functions approximation.** For $y \in \mathbb{R}$ and $B > 0$ we denote $[y]_{[-B,B]} = \max(-B, \min(y, B))$, namely, clipping $y$ to the interval $[-B, B]$. We say that $f$ is *approximately* $\mathrm{poly}(d)$-*bounded* if for every $\epsilon = \frac{1}{\mathrm{poly}(d)}$ there is $B = \mathrm{poly}(d)$ such that $\mathbb{E}_{\mathbf{x} \sim \mathcal{D}}\left(f(\mathbf{x}) - [f(\mathbf{x})]_{[-B,B]}\right)^2 \leq \epsilon$. Note that if $f$ is bounded by some $B = \mathrm{poly}(d)$ then it is also approximately $\mathrm{poly}(d)$-bounded. We say that $f$ can be *approximated by a neural network of depth $k$* (with respect to a distribution $\mathcal{D}$) if for every $\epsilon = \frac{1}{\mathrm{poly}(d)}$ we have $\mathbb{E}_{\mathbf{x} \sim \mathcal{D}}(f(\mathbf{x}) - N(\mathbf{x}))^2 \leq \epsilon$ for some depth-$k$ network $N$ of width (and size) $\mathrm{poly}(d)$.

**Depth separation.** We say that there is depth-separation between networks of depth $k$ and depth $k'$ for some integers $k' > k$, if there is a distribution $\mathcal{D}$ on $\mathbb{R}^d$ and a function $f : \mathbb{R}^d \to \mathbb{R}$ that can be approximated (with respect to $\mathcal{D}$) by a neural network of depth $k'$ but cannot be approximated by a network of depth $k$. We note that our definition of depth-separation is a bit weaker than most existing depth-separation results, which actually show difficulty of approximation even up to constant accuracy (and not just $1/\mathrm{poly}(d)$ accuracy). However, depth separation in that sense implies depth separation in our sense. Hence, the barriers we show here for depth separation imply similar barriers under this other (or any stronger) notion of depth separation.

# 3 Results

We start by presenting our results on small-weight networks, implying that extremely large weights do not significantly help neural networks to express approximately $\mathrm{poly}(d)$-bounded functions. We show this via a positive result: If an approximately $\mathrm{poly}(d)$-bounded function can be approximated by a network of constant depth $k$, then it can also be approximated by a depth-$(3k+3)$ network with $\mathrm{poly}(d)$-bounded weights. We then proceed to establish depth-separation barriers.

## 3.1 Neural networks with small weights

We start with the case where the input distribution is continuous:

**Theorem 3.1.** *Let $\mu$ be a density function on $\mathbb{R}^d$ with an almost-bounded support and almost-bounded conditional density. Let $f : \mathbb{R}^d \to \mathbb{R}$ be an approximately $\mathrm{poly}(d)$-bounded function, and let $k$ be a constant, namely, independent of $d$. If $f$ can be approximated by a neural network of depth $k$ and width $\mathrm{poly}(d)$, then it can also be approximated by a neural network of depth $3k + 3$, width $\mathrm{poly}(d)$, and $\mathrm{poly}(d)$-bounded weights.*

We now show a similar result for the case where the input distribution is discrete:

**Theorem 3.2.** *Let $R(d)$ and $p(d)$ be any polynomials in $d$, and let $\mathcal{I} = \{\frac{j}{p} : -R \cdot p \leq j \leq R \cdot p, j \in \mathbb{Z}\}$. Let $\mathcal{D}$ be a distribution on $\mathcal{I}^d$. Let $f : \mathbb{R}^d \to \mathbb{R}$ be an approximately $\mathrm{poly}(d)$-bounded function, and let $k$ be a constant, namely, independent of $d$. If $f$ can be approximated by a neural network of depth $k$ and width $\mathrm{poly}(d)$, then it can also be approximated by a neural network of depth $3k + 3$, width $\mathrm{poly}(d)$, and $\mathrm{poly}(d)$-bounded weights.*

**Remark 3.1.** *Since we require $\mathrm{poly}(d)$ width, then Theorems 3.1 and 3.2 imply that $f$ can also be approximated by a network of depth $3k+3$ with constant weights (at the expense of a $\mathrm{poly}(d)$ blowup in the width, by recursively substituting every neuron with $\mathrm{poly}(d)$-many constant-weight neurons).*

**Remark 3.2.** *Note that from Theorems 3.1 and 3.2, it follows that under the assumptions stated there, any $\mathrm{poly}(d)$-bounded function that can be approximated by a constant-depth network can also be approximated by a constant-depth network that is $\mathrm{poly}(d)$-Lipschitz.*

**Remark 3.3.** *In Theorems 3.1 and 3.2, we obtain a network of depth $3k + 3$, width $\mathrm{poly}(d)$, and $\mathrm{poly}(d)$-bounded weights. Since $k$ is a constant, we hide the dependence of the width and of the weights on $k$ inside the $poly()$ notation. We note though that this dependence is exponential in $k$.*

## 3.2 Barriers to depth separation

The following theorem enables us to leverage known barriers to depth separation for threshold circuits in order to obtain barriers to depth separation for neural networks.

**Theorem 3.3.** *Let $\mu$ be a density function on $\mathbb{R}^d$ with an almost-bounded support and almost-bounded conditional density. Let $f : \mathbb{R}^d \to \mathbb{R}$ be an approximately $\mathrm{poly}(d)$-bounded function, and let $k' > k \geq 4$ be constants. If $f$ cannot be approximated by a neural network of depth $k$ and width $\mathrm{poly}(d)$, but can be approximated by a neural network of depth $k'$ and width $\mathrm{poly}(d)$, then there is a function $g : \{0,1\}^{d'} \to \{0,1\}$ that cannot be computed by a $\mathrm{poly}(d')$-sized threshold circuit of depth $k - 2$, but can be computed by a $\mathrm{poly}(d')$-sized threshold circuit of depth $3k' + 1$.*

The main focus in the existing works on depth-separation in neural networks is on continuous input distributions. However, it is also important to study the case where the input distribution is discrete. In the following theorem we show that the barriers to depth separation also hold in this case.

**Theorem 3.4.** *Let $R(d)$ and $p(d)$ be any polynomials in $d$, and let $\mathcal{I} = \{\frac{j}{p} : -Rp \leq j \leq Rp, j \in \mathbb{Z}\}$. Let $\mathcal{D}$ be a distribution on $\mathcal{I}^d$. Let $f : \mathbb{R}^d \to \mathbb{R}$ be an approximately $\mathrm{poly}(d)$-bounded function, and let $k' > k \geq 4$ be constants. If $f$ cannot be approximated by a neural network of width $\mathrm{poly}(d)$ and depth $k$, but can be approximated by a network of width $\mathrm{poly}(d)$ and depth $k'$, then there is a function $g : \{0,1\}^{d'} \to \{0,1\}$ that cannot be computed by a $\mathrm{poly}(d')$-sized threshold circuit of depth $k - 2$, but can be computed by a $\mathrm{poly}(d')$-sized threshold circuit of depth $3k' + 1$.*

**Remark 3.4.** *From Theorems 3.3 and 3.4, it follows that depth-separation between neural networks of depth $k \geq 4$ and some constant $k' > k$, would imply depth separation between threshold circuits of depth $k - 2$ and some constant greater than $k - 2$. Hence, showing depth separation with $k = 5$ would solve the longstanding open problem of separating between threshold circuits of depth $3$ and some larger constant. Showing depth separation with $k \geq 6$ would solve the open problem of separating between circuits of depth $k - 2$ and some larger constant, which is especially challenging due to the natural-proof barrier for threshold circuits. Finally, showing depth separation with $k = 4$ would solve the longstanding open problem of separating between circuits of depth $2$ (with arbitrarily large weights) and some larger constant. Recall that separation between threshold circuits of depth $2$ and $3$ is known only under the assumption that the weight magnitudes are $\mathrm{poly}(d)$ bounded.*

**Remark 3.5.** *Sometimes when considering depth separation in neural networks, it is useful to restrict the magnitude of the weights. For example, [6] gave a function that can be approximated by a depth-3 network of $\mathrm{poly}(d)$ width and $\mathrm{poly}(d)$-bounded weights, but cannot be approximated by a depth-2 network of $\mathrm{poly}(d)$ width and weights bounded by $2^d$. We note that our barrier applies also to this type of separation. Namely, depth-separation for neural networks of $\mathrm{poly}(d)$-bounded weights between depth $k$ and some constant $k' > k$, would imply depth-separation for threshold circuits of $\mathrm{poly}(d)$-bounded weights between depth $k-2$ and some constant greater than $k-2$. Such separation for threshold circuits is an open problem for circuits of depth at least 3 ([22]), and has a natural-proof barrier for circuits of depth at least 4 ([16]).*

While Theorems 3.3 and 3.4 give a strong barrier to depth separation, it should not discourage researchers from continuing to investigate the problem, as discussed in Section 1. Moreover, our barrier does not apply to separation between depth 3 and some larger constant. However, we now show that even for this case, a depth-separation result would require some different approach than these used in existing results. As we discussed in Section 1, in the existing depth-separation results for continuous input distributions, $f$ is either a radial function or a function that depends only on one component. In the following theorems we formally show that for such functions, a network of a constant depth greater than 3 does not have more power than a network of depth 3 (we note that similar results appeared in e.g., [7, 6] in the context of specific radial functions, and we actually rely on a technical lemma presented by the former reference).

**Theorem 3.5.** *Let $\mu$ be a distribution on $\mathbb{R}^d$ with an almost-bounded support and almost-bounded conditional density. Let $f : \mathbb{R}^d \to \mathbb{R}$ be an approximately $\mathrm{poly}(d)$-bounded function, that can be approximated by a neural network of $\mathrm{poly}(d)$ width and constant depth. If $f$ and $\mu$ are radial, then $f$ can be approximated by a network of width $\mathrm{poly}(d)$, depth 3, and $\mathrm{poly}(d)$-bounded weights.*

**Theorem 3.6.** *Let $\mu$ be a distribution on $\mathbb{R}^d$ such that the $d$ components are drawn independently. Let $f : \mathbb{R}^d \to \mathbb{R}$ be a function that can be approximated by a neural network of $\mathrm{poly}(d)$ width and constant depth. If $f(\mathbf{x}) = \sum_{i \in [d]} f_i(x_i)$ for functions $f_i : \mathbb{R} \to \mathbb{R}$, then $f$ can be approximated by a network of width $\mathrm{poly}(d)$ and depth 2.*

## 4 Proof ideas

In this section we describe the main ideas of the proofs of Theorems 3.1, 3.2, 3.3 and 3.4.

### 4.1 Neural networks with small weights

Let $\epsilon = \frac{1}{\mathrm{poly}(d)}$. For simplicity, we assume that there is a network $N'$ of constant depth and $\mathrm{poly}(d)$ width, such that for some $B = \mathrm{poly}(d)$ we have for every $\mathbf{x}$ that $N'(\mathbf{x}) \in [-B, B]$, and $\|N' - f\|_{L_2(\mathcal{D})} \le \frac{\epsilon}{2}$ (or $\|N' - f\|_{L_2(\mu)} \le \frac{\epsilon}{2}$, for a continuous distribution). Thus, $N'$ approximates $f$ and is bounded by $B$. We will construct a network $\hat{N}$ of constant depth, $\mathrm{poly}(d)$-width and $\mathrm{poly}(d)$-bounded weights, such that $\|\hat{N} - N'\|_{L_2(\mathcal{D})} \le \frac{\epsilon}{2}$ (respectively, $\|\hat{N} - N'\|_{L_2(\mu)} \le \frac{\epsilon}{2}$).

#### 4.1.1 Discrete input distributions

First, we show that for every polynomial $p'(d)$, we can construct a network $N''$ of constant depth and $\mathrm{poly}(d)$ width, such that for every $\mathbf{x} \in \mathcal{I}^d$ we have $|N''(\mathbf{x}) - N'(\mathbf{x})| \le \frac{1}{p'(d)}$, and there exists a positive integer $t \le 2^{\mathrm{poly}(d)}$ such that all weights and biases in $N''$ are in $Q_t = \{\frac{s}{t} : |s| \le 2^{\mathrm{poly}(d)}, s \in \mathbb{Z}\}$. Thus, for a sufficiently large polynomial $p'(d)$, we have $\|N'' - N'\|_{L_2(\mathcal{D})} \le \frac{\epsilon}{2}$, and all weights and biases in $N''$ can be represented by $\mathrm{poly}(d)$ bits. The idea of the construction of $N''$ is as follows. First, we transform $N'$ into a network $\mathcal{N}$ with a special structure (and arbitrary weights) that computes the same function. Then, we define a (very large) system of linear inequalities, such that for every $\mathbf{x} \in \mathcal{I}^d$ we have inequalities that correspond to the computation $\mathcal{N}(\mathbf{x})$. The variables in the linear system correspond to the weights and biases in $\mathcal{N}$. Finally, we show that the system has a solution such that all values are in $Q_t$ for some positive integer $t \le 2^{\mathrm{poly}(d)}$, and that this solution induces a network $N''$ where $|N''(\mathbf{x}) - N'(\mathbf{x})| \le \frac{1}{p'(d)}$ for every $\mathbf{x} \in \mathcal{I}^d$. We note that a similar idea was used in [18]. However, we use a different construction, since we consider approximation of real-valued functions, while that paper considered exact computation of Boolean functions.

To further reduce the weight magnitudes from $2^{\text{poly}(d)}$ to $\text{poly}(d)$, we note that $N''(\mathbf{x})$ can be simulated by representing all inputs, internal values and weights of $N''$ by binary vectors, and computing each layer by applying arithmetic operations on the binary vectors. We construct a new network $\hat{N}$ as follows: First, it transforms the input $\mathbf{x} \in \mathcal{I}^d$ to a binary representation. Since for every $i \in [d]$ the component $x_i$ is of the form $\frac{j}{p(d)}$ for some integer $-Rp(d) \leq j \leq Rp(d)$, and since $R, p(d)$ are polynomials, then a binary representation of $\mathbf{x}$ can be computed by two layers of width $\text{poly}(d)$ with $\text{poly}(d)$-bounded weights. Second, it simulates $N''(\mathbf{x})$ using arithmetic operations on binary vectors. We show that each such operation can be done by a constant number of layers with $\text{poly}(d)$ width and $\text{poly}(d)$-bounded weights. Finally, since $N''(\mathbf{x})$ is $\text{poly}(d)$-bounded, it can be transformed from a binary representation to its real value while using $\text{poly}(d)$-bounded weights.

### 4.1.2 Continuous input distributions

In Section 4.1.1, we described how to approximate a network $N'$ with arbitrary weights by a network $\hat{N}$ with small weights, where the inputs are discrete. In order to handle continuous input distributions, we will first "round" the input, namely, transform an input $\mathbf{x}$ to the nearest point $\tilde{\mathbf{x}}$ in some discrete set. Then, we will use the construction from Section 4.1.1 in order to approximate $N'(\tilde{\mathbf{x}})$. Note that we do not have any guarantees regarding the Lipschitzness of $N'$, and therefore it is possible that $|N'(\mathbf{x}) - N'(\tilde{\mathbf{x}})|$ is large. Thus, it is not obvious that such a construction approximates $N'$. However, we will show that even though $N'$ is not Lipschitz, $|N'(\mathbf{x}) - N'(\tilde{\mathbf{x}})|$ is small w.h.p. over $\mathbf{x}$. Intuitively, the reason is that $N'$ is a bounded function, and has a piecewise-linear structure with a bounded number of pieces along a path. Thus, the measure of the linear segments with a huge Lipschitz constant cannot be too large. Therefore, if we sample $\mathbf{x}$ and then move from $\mathbf{x}$ to $\tilde{\mathbf{x}}$, the probability that we cross an interval with a huge Lipschitz constant is small.

We now turn to describe the proof ideas in slightly more technical detail. For simplicity, instead of assuming that $\mu$ has an almost-bounded support, we assume that its support is contained in $[-R, R]^d$ for $R = \text{poly}(d)$. Let $p(d)$ be a polynomial and let $\mathcal{I} = \{\frac{j}{p(d)} : -Rp(d) \leq j \leq Rp(d), j \in \mathbb{Z}\}$. Let $\mathbf{x} \in [-R, R]^d$. For $i \in [d]$, let $\tilde{x}_i \in \mathcal{I}$ be such that $|\tilde{x}_i - x_i|$ is minimal. That is, $\tilde{x}_i$ is obtained by rounding $x_i$ to the nearest multiple of $\frac{1}{p(d)}$. Let $\tilde{\mathbf{x}} = (\tilde{x}_1, \ldots, \tilde{x}_d)$. Let $\tilde{N} : \mathbb{R}^d \to [-B, B]$ be a function such that for every $\mathbf{x} \in [-R, R]^d$ we have $\tilde{N}(\mathbf{x}) = N'(\tilde{\mathbf{x}})$. We will show that $\|\tilde{N} - N'\|_{L_2(\mu)} \leq \frac{\epsilon}{4}$, and then construct a network $\hat{N}$ of constant depth, $\text{poly}(d)$-width and $\text{poly}(d)$-bounded weights, such that $\|\hat{N} - \tilde{N}\|_{L_2(\mu)} \leq \frac{\epsilon}{4}$. It implies that $\|\hat{N} - N'\|_{L_2(\mu)} \leq \frac{\epsilon}{2}$ as required.

We start with the main idea in the proof of $\|\tilde{N} - N'\|_{L_2(\mu)} \leq \frac{\epsilon}{4}$. Since $N'$ is bounded by $B$, then for every $\mathbf{x} \in [-R, R]^d$ we have $|\tilde{N}(\mathbf{x}) - N'(\mathbf{x})| \leq 2B$. In order to bound $\|\tilde{N} - N'\|_{L_2(\mu)}$ we need to show that w.h.p. $|\tilde{N}(\mathbf{x}) - N'(\mathbf{x})|$ is small. Namely, that w.h.p. the value of $N'$ does not change too much by moving from $\mathbf{x}$ to $\tilde{\mathbf{x}}$. Intuitively, it follows from the following argument. We move from $\mathbf{x}$ to $\tilde{\mathbf{x}}$ in $d$ steps. In the $i$-th step we change the $i$-th component from $x_i$ to $\tilde{x}_i$. Namely, we move from $(\tilde{x}_1, \ldots, \tilde{x}_{i-1}, x_i, x_{i+1}, \ldots, x_d)$ to $(\tilde{x}_1, \ldots, \tilde{x}_{i-1}, \tilde{x}_i, x_{i+1}, \ldots, x_d)$. We show that in each step, w.h.p., the change in $N'$ is small. Since in the $i$-th step the components $[d] \setminus \{i\}$ are fixed, then the dependence of $N'$ on the value of the $i$-th component, which is the component that we change, can be expressed by a network with input dimension 1, width $\text{poly}(d)$, and constant depth. Such a network computes a function $g_i : \mathbb{R} \to \mathbb{R}$ that is piecewise linear with $\text{poly}(d)$ pieces. Since $N'$ is bounded by $B$ then $g_i$ is also bounded by $B$, and therefore a linear piece in $g_i$ whose derivative has a large absolute value, is supported on a small interval. Now, we are able to show that w.h.p. the interval between $x_i$ and $\tilde{x}_i$ has an empty intersection with intervals of $g_i$ whose derivatives have large absolute values. Hence, w.h.p. the change in $N'$ in the $i$-th step is small.

We now describe the network $\hat{N}$. First, $\hat{N}$ transforms w.h.p. the input $\mathbf{x}$ into $\tilde{\mathbf{x}}$. Note that the mapping $\mathbf{x} \mapsto \tilde{\mathbf{x}}$ is not continuous and hence cannot be computed by a neural network for all $\mathbf{x} \in [-R, R]^d$, but it can be done w.h.p. where $\mathbf{x} \sim \mu$, by two layers of width $\text{poly}(d)$ and $\text{poly}(d)$-bounded weights. Then, using the construction described in Section 4.1.1, the network $\hat{N}$ approximates $N'(\tilde{\mathbf{x}})$.

## 4.2 Barriers to depth separation

Let $N$ be a network of depth $k'$ that approximates $f$. In Section 4.1 we constructed a network $\hat{N}$ with small weights that approximates $f$. We show that the network $\hat{N}$ is of depth $3k' + 3$, and that layers $2, \ldots, 3k' + 2$ in $\hat{N}$ can be expressed by a threshold circuit $T$. Let $g : \{0, 1\}^{d'} \to \{0, 1\}$ be the function that $T$ computes. Assume that $g$ can be computed by a threshold circuit $T'$ of depth $k - 2$. Now, we can replace the layers in $\hat{N}$ that compute $T$ by layers that simulate $T'$, and obtain a network of depth $k$ that approximates $f$, in contradiction to the assumption.

## Broader impact

Not applicable as far as we can see (this is a purely theoretical paper).

## Funding disclosure

This research is supported in part by European Research Council (ERC) grant 754705.

## Footnotes

[1]Note that in this literature it is customary to require exact representation of the function, rather than merely approximating it.

[2]In [6] the function is not radial, but, as shown in [25], it can be reduced to a radial one.

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
