[Supplementary Material]

# A Proofs

## A.1 Proof of Theorem 3.1

Let $\epsilon = \frac{1}{\text{poly}(d)}$. Let $N$ be a neural network of depth $k$ and width $\text{poly}(d)$, such that $\|N - f\|_{L_2(\mu)} \leq \frac{\epsilon}{5}$. We will construct a network $\hat{N}$ of depth $3k + 3$, width $\text{poly}(d)$ and $\text{poly}(d)$-bounded weights, such that $\|\hat{N} - f\|_{L_2(\mu)} \leq \epsilon$.

Since $f$ is approximately $\text{poly}(d)$-bounded, there is $B = \text{poly}(d)$ be such that

$$\mathbb{E}_{\mathbf{x} \sim \mu} \left( f(\mathbf{x}) - [f(\mathbf{x})]_{[-B,B]} \right)^2 \leq \left( \frac{\epsilon}{5} \right)^2 .$$

Let $f' : \mathbb{R}^d \to \mathbb{R}$ be such that $f'(\mathbf{x}) = [f(\mathbf{x})]_{[-B,B]}$. Thus,

$$\|f' - f\|_{L_2(\mu)} \leq \frac{\epsilon}{5} . \tag{1}$$

Let $N'$ be a network of depth $k+1$ such that for every $\mathbf{x} \in \mathbb{R}^d$ we have $N'(\mathbf{x}) = [N(\mathbf{x})]_{[-B,B]}$. Such $N'$ can be obtained from $N$ by adding to it one layer, since $N'(\mathbf{x}) = [N(\mathbf{x})+B]_+ - [N(\mathbf{x})-B]_+ - B$. Although we do not allow bias in the output neuron, the additive term $-B$ can be implemented by adding a hidden neuron with fan-in 0 and bias 1, that is connected to the output neuron with weight $-B$. Note that $\mathbb{E}_{\mathbf{x} \sim \mu}(N'(\mathbf{x}) - f'(\mathbf{x}))^2 \leq \mathbb{E}_{\mathbf{x} \sim \mu}(N(\mathbf{x}) - f'(\mathbf{x}))^2$, and therefore

$$\|N' - f'\|_{L_2(\mu)} \leq \|N - f'\|_{L_2(\mu)} \leq \|N - f\|_{L_2(\mu)} + \|f - f'\|_{L_2(\mu)} \leq \frac{2\epsilon}{5} . \tag{2}$$

Let $\delta = \frac{\epsilon^2}{400 B^2}$. Since $\mu$ has an almost-bounded support, there is $R = \text{poly}(d)$ such that $Pr_{\mathbf{x} \sim \mu}(\mathbf{x} \notin [-R,R]^d) \leq \delta$. Let $p(d)$ be a polynomial. Let $\mathcal{I} = \{ \frac{j}{p(d)} : -Rp(d) \leq j \leq Rp(d), j \in \mathbb{Z} \}$. Let $\mathbf{x} \in \mathbb{R}^d$. For every $i$ such that $x_i \in [-R - \frac{1}{2p(d)}, R + \frac{1}{2p(d)}]$, let $\tilde{x}_i \in \mathcal{I}$ be such that $|\tilde{x}_i - x_i|$ is minimal. That is, $\tilde{x}_i$ is obtained by rounding $x_i$ to the nearest multiple of $\frac{1}{p(d)}$. For every $i$ such that $x_i \notin [-R - \frac{1}{2p(d)}, R + \frac{1}{2p(d)}]$, let $\tilde{x}_i = 0$. Then, let $\tilde{\mathbf{x}} = (\tilde{x}_1, \ldots, \tilde{x}_d)$. Let $\tilde{N} : \mathbb{R}^d \to [-B, B]$ be a function such that for every $\mathbf{x} \in \mathbb{R}^d$ we have $\tilde{N}(\mathbf{x}) = N'(\tilde{\mathbf{x}})$. We will prove the following two lemmas:

**Lemma A.1.** *There exists a polynomial $p(d)$ such that*

$$\|\tilde{N} - N'\|_{L_2(\mu)} \leq \frac{\epsilon}{5} .$$

**Lemma A.2.** *There exists a neural network $\hat{N}$ of depth $3k+3$, width $\text{poly}(d)$ and $\text{poly}(d)$-bounded weights, such that*

$$\|\hat{N} - \tilde{N}\|_{L_2(\mu)} \leq \frac{\epsilon}{5} .$$

Then, combining Lemmas A.1 and A.2 with Eq. 1 and 2, we have

$$\|\hat{N} - f\|_{L_2(\mu)} \leq \|\hat{N} - \tilde{N}\|_{L_2(\mu)} + \|\tilde{N} - N'\|_{L_2(\mu)} + \|N' - f'\|_{L_2(\mu)} + \|f' - f\|_{L_2(\mu)}$$

$$\leq \frac{\epsilon}{5} + \frac{\epsilon}{5} + \frac{2\epsilon}{5} + \frac{\epsilon}{5} = \epsilon ,$$

and hence the theorem follows.

### A.1.1 Proof of Lemma A.1

We start with an intuitive explanation, and then turn to the formal proof. Since we have $\tilde{N}(\mathbf{x}) = N'(\tilde{\mathbf{x}})$ and $|N'(\mathbf{x})| \leq B$ for every $\mathbf{x}$, then we have $|\tilde{N}(\mathbf{x}) - N'(\mathbf{x})| \leq 2B$. In order to bound $\|\tilde{N} - N'\|_{L_2(\mu)}$ we show that w.h.p. $|\tilde{N}(\mathbf{x}) - N'(\mathbf{x})|$ is small. Namely, that w.h.p. the value of $N'$ does not change too much by moving from $\mathbf{x}$ to $\tilde{\mathbf{x}}$. Since the Lipschitzness of $N'$ is not bounded, then for every choice of a polynomial $p(d)$, we are not guaranteed that $|N'(\tilde{\mathbf{x}}) - N'(\mathbf{x})|$ is small. However, we show that for a sufficiently large polynomial $p(d)$, the probability that we encounter a region where $N'$ has large derivative while moving from $\mathbf{x}$ to $\tilde{\mathbf{x}}$, is small.

We move from $\mathbf{x}$ to $\tilde{\mathbf{x}}$ in $d$ steps. In the $i$-th step we change the $i$-th component from $x_i$ to $\tilde{x}_i$. Namely, we move from $(\tilde{x}_1, \ldots, \tilde{x}_{i-1}, x_i, x_{i+1}, \ldots, x_d)$ to $(\tilde{x}_1, \ldots, \tilde{x}_{i-1}, \tilde{x}_i, x_{i+1}, \ldots, x_d)$. We show that for each step, w.h.p., the change in $N'$ is small. Since in the $i$-th step the components $[d] \setminus \{i\}$ are fixed, then the dependence of $N'$ on the value of the $i$-th component, which is the component that we change, can be expressed by a network with input dimension 1, width $\text{poly}(d)$, and constant depth. Such a network computes a function $g_i : \mathbb{R} \to \mathbb{R}$ that is piecewise linear with $\text{poly}(d)$ pieces. Since $N'$ is bounded by $B$ then $g_i$ is also bounded by $B$, and therefore a linear piece in $g_i$ whose derivative has a large absolute value is supported on a small interval. Now, we need to show that the interval between $x_i$ and $\tilde{x}_i$ has an empty intersection with intervals of $g_i$ with large derivatives. Since there are only $\text{poly}(d)$ intervals and intervals with large derivatives are small, then by using the fact that $\mu$ has an almost-bounded conditional density, we are able to show that w.h.p. the interval between $x_i$ and $\tilde{x}_i$ does not have a non-empty intersection with such intervals. Intuitively, we can think about the choice of $\mathbf{x} \sim \mu$ as choosing the components $[d] \setminus \{i\}$ according to $\mu_{[d] \setminus i}$ and then choosing $x_i$ according to $\mu_{i | [d] \setminus i}$. Now, the choice of the components $[d] \setminus \{i\}$ induces the function $g_i$, and the choice of $x_i$ is good with respect to $g_i$ if the interval between $x_i$ and $\tilde{x}_i$ does not have a non-empty intersection with the intervals of $g_i$ which have large derivatives. We show that w.h.p. we obtain $g_i$ and $x_i$, such that $x_i$ is good with respect to $g_i$.

We now turn to the formal proof. Let

$$A = \left\{ \mathbf{x} \in \mathbb{R}^d : (N'(\mathbf{x}) - \tilde{N}(\mathbf{x}))^2 > \frac{\epsilon^2}{50} \right\} .$$

Let $\mathbf{x} \in \mathbb{R}^d$, let $\mathbf{y}_i = (\tilde{x}_1, \ldots, \tilde{x}_{i-1}, x_{i+1}, \ldots, x_d) \in \mathbb{R}^{d-1}$, and let $N'_{i, \mathbf{y}_i} : \mathbb{R} \to [-B, B]$ be such that

$$N'_{i, \mathbf{y}_i}(t) = N'(\tilde{x}_1, \ldots, \tilde{x}_{i-1}, t, \ldots, x_d) .$$

Note that for every $\mathbf{x} \in \mathbb{R}^d$, we have

$$|N'(\mathbf{x}) - \tilde{N}(\mathbf{x})| = |N'(\mathbf{x}) - N'(\tilde{\mathbf{x}})|$$

$$= \left| \sum_{i \in [d]} N'(\tilde{x}_1, \ldots, \tilde{x}_{i-1}, x_i, \ldots, x_d) - N'(\tilde{x}_1, \ldots, \tilde{x}_i, x_{i+1}, \ldots, x_d) \right|$$

$$\leq \sum_{i \in [d]} |N'(\tilde{x}_1, \ldots, \tilde{x}_{i-1}, x_i, \ldots, x_d) - N'(\tilde{x}_1, \ldots, \tilde{x}_i, x_{i+1}, \ldots, x_d)|$$

$$= \sum_{i \in [d]} |N'_{i, \mathbf{y}_i}(x_i) - N'_{i, \mathbf{y}_i}(\tilde{x}_i)| .$$

Thus, using the shorthand $Pr(\cdot \mid [-R, R]^d)$ for $Pr(\cdot \mid \mathbf{x} \in [-R, R]^d)$, we have

$$Pr(A \mid \mathbf{x} \in [-R, R]^d) = Pr\left( (N'(\mathbf{x}) - \tilde{N}(\mathbf{x}))^2 > \frac{\epsilon^2}{50} \Big| [-R, R]^d \right)$$

$$= Pr\left( |N'(\mathbf{x}) - \tilde{N}(\mathbf{x})| > \frac{\epsilon}{5\sqrt{2}} \Big| [-R, R]^d \right)$$

$$\leq Pr\left( \sum_{i \in [d]} |N'_{i, \mathbf{y}_i}(x_i) - N'_{i, \mathbf{y}_i}(\tilde{x}_i)| > \frac{\epsilon}{5\sqrt{2}} \Big| [-R, R]^d \right)$$

$$\leq Pr\left( \exists i \in [d] \text{ s.t. } |N'_{i, \mathbf{y}_i}(x_i) - N'_{i, \mathbf{y}_i}(\tilde{x}_i)| > \frac{\epsilon}{5\sqrt{2}d} \Big| [-R, R]^d \right)$$

$$\leq \sum_{i \in [d]} Pr\left( |N'_{i, \mathbf{y}_i}(x_i) - N'_{i, \mathbf{y}_i}(\tilde{x}_i)| > \frac{\epsilon}{5\sqrt{2}d} \Big| [-R, R]^d \right) .$$

Now, since $Pr(\mathbf{x} \notin [-R, R]^d) \leq \delta$, we have

$$Pr(A) = Pr(A \mid [-R, R]^d) \cdot Pr([-R, R]^d) + Pr(A \mid \mathbb{R}^d \setminus [-R, R]^d) \cdot Pr(\mathbb{R}^d \setminus [-R, R]^d)$$

$$\leq \sum_{i \in [d]} Pr\left( |N'_{i, \mathbf{y}_i}(x_i) - N'_{i, \mathbf{y}_i}(\tilde{x}_i)| > \frac{\epsilon}{5\sqrt{2}d} \Big| [-R, R]^d \right) + \delta . \tag{3}$$

Let
$$A_i = \left\{ \mathbf{x} \in \mathbb{R}^d : |N'_{i,\mathbf{y}_i}(x_i) - N'_{i,\mathbf{y}_i}(\tilde{x}_i)| > \frac{\epsilon}{5\sqrt{2}d} \right\} .$$

**Lemma A.3.**
$$Pr\left(A_i \mid \mathbf{x} \in [-R, R]^d\right) \leq \frac{\epsilon^2}{400B^2 d} .$$

*Proof.* Let $\delta' = \frac{\epsilon^2}{1600B^2 d}$. Since $\mu$ has an almost-bounded conditional density, there is $M = \text{poly}(d)$ such that for every $i \in [d]$ we have $Pr(G_i) \leq \delta'$, where

$$G_i = \{\mathbf{x} \in \mathbb{R}^d : \exists t \in \mathbb{R} \text{ s.t. } \mu_{i|[d]\setminus i}(t|x_1, \ldots, x_{i-1}, x_{i+1}, \ldots, x_d) > M\} .$$

Now, we have

$$
\begin{aligned}
Pr\left(A_i|[-R,R]^d\right) &= Pr\left(A_i \cap G_i|[-R,R]^d\right) + Pr\left(A_i \cap (\mathbb{R}^d \setminus G_i)|[-R,R]^d\right) \\
&\leq Pr\left(G_i|[-R,R]^d\right) + Pr\left(A_i \cap (\mathbb{R}^d \setminus G_i)|[-R,R]^d\right)
\end{aligned}
\tag{4}
$$

Note that

$$Pr\left(G_i|[-R,R]^d\right) = \frac{Pr\left(G_i \cap [-R,R]^d\right)}{Pr\left([-R,R]^d\right)} \leq \frac{Pr(G_i)}{Pr\left([-R,R]^d\right)} \leq \frac{\delta'}{1 - \delta} . \tag{5}$$

Thus, it remains to bound $Pr\left(A_i \cap (\mathbb{R}^d \setminus G_i)|[-R,R]^d\right)$. Let $\mathbf{x} \in [-R, R]^d$. Note that the function $N'_{i,\mathbf{y}_i} : \mathbb{R} \to \mathbb{R}$ can be expressed by a neural network of depth $k+1$ that is obtained from $N'$ by using the hardwired $\mathbf{y}_i$ instead of the corresponding input components $[d] \setminus \{i\}$. That is, if a neuron in the first hidden layer of $N'$ has weights $w_1, \ldots, w_d$ and bias $b$, then in $N'_{i,\mathbf{y}_i}$ its weight is $w_i$ and its bias is $b + \langle(w_1, \ldots, w_{i-1}, w_{i+1}, \ldots, w_d), \mathbf{y}_i\rangle$. A neural network with input dimension 1, constant depth, and $m$ neurons in each hidden layer, is piecewise linear with at most $\text{poly}(m)$ pieces ([29]). Therefore, $N'_{i,\mathbf{y}_i}$ consists of $l = \text{poly}(d)$ linear pieces. Note that $l$ depends only on the depth and width of $N'$, and does not depend on $i$ and $\mathbf{y}_i$.

Since $N'_{i,\mathbf{y}_i}(t) \in [-B, B]$ for every $t \in \mathbb{R}$, then if $N'_{i,\mathbf{y}_i}$ has derivative $\alpha$ in a linear interval $[a, b]$ then $|(b-a)\alpha| \leq 2B$. Let $\mathbf{x}^i = (x_1, \ldots, x_{i-1}, x_{i+1}, \ldots, x_d) \in \mathbb{R}^{d-1}$. Let $\gamma = \frac{6400B^3 dMl}{\epsilon^2}$. We denote by $I_{i,\mathbf{x}^i,\gamma}$ the set of intervals $[a_j, b_j]$ where the derivative $\alpha_j$ in $N'_{i,\mathbf{y}_i}$ satisfies $|\alpha_j| > \gamma$. Note that $\mathbf{y}_i$ depends on $\mathbf{x}^i$ and does not depend on $x_i$. Now,

$$\sum_{[a_j,b_j]\in I_{i,\mathbf{x}^i,\gamma}} (b_j - a_j) \leq \sum_{[a_j,b_j]\in I_{i,\mathbf{x}^i,\gamma}} \frac{2B}{|\alpha_j|} < l \cdot \frac{2B}{\gamma} . \tag{6}$$

Let $\beta$ be the open interval $(x_i, \tilde{x}_i)$ if $x_i \leq \tilde{x}_i$ or $(\tilde{x}_i, x_i)$ otherwise. If $\beta \cap [a_j, b_j] = \emptyset$ for every $[a_j, b_j] \in I_{i,\mathbf{x}^i,\gamma}$, then

$$|N'_{i,\mathbf{y}_i}(x_i) - N'_{i,\mathbf{y}_i}(\tilde{x}_i)| \leq |\tilde{x}_i - x_i|\gamma \leq \frac{\gamma}{p(d)} .$$

Let

$$I'_{i,\mathbf{x}^i,\gamma} = \left\{ [a'_j, b'_j] : a'_j = a_j - \frac{1}{p(d)}, b'_j = b_j + \frac{1}{p(d)}, [a_j, b_j] \in I_{i,\mathbf{x}^i,\gamma} \right\} .$$

Thus, if $|N'_{i,\mathbf{y}_i}(x_i) - N'_{i,\mathbf{y}_i}(\tilde{x}_i)| > \frac{\gamma}{p(d)}$ then $\beta \cap [a_j, b_j] \neq \emptyset$ for some $[a_j, b_j] \in I_{i,\mathbf{x}^i,\gamma}$, and therefore $x_i \in [a'_j, b'_j]$ for some $[a'_j, b'_j] \in I'_{i,\mathbf{x}^i,\gamma}$. Hence, for a sufficiently large polynomial $p(d)$, if $\mathbf{x} \in A_i$ then $x_i \in [a'_j, b'_j]$ for some $[a'_j, b'_j] \in I'_{i,\mathbf{x}^i,\gamma}$.

We denote $\ell(I'_{i,\mathbf{x}^i,\gamma}) = \sum_{[a'_j,b'_j]\in I'_{i,\mathbf{x}^i,\gamma}} (b'_j - a'_j)$. Note that

$$\ell(I'_{i,\mathbf{x}^i,\gamma}) = \sum_{[a_j,b_j]\in I_{i,\mathbf{x}^i,\gamma}} (b_j - a_j + \frac{2}{p(d)}) \leq \frac{2l}{p(d)} + \sum_{[a_j,b_j]\in I_{i,\mathbf{x}^i,\gamma}} (b_j - a_j) \overset{(Eq.\ 6)}{<} \frac{2l}{p(d)} + \frac{2Bl}{\gamma} . \tag{7}$$

For $\mathbf{z} \in \mathbb{R}^{d-1}$ and $t \in \mathbb{R}$ we denote $\mathbf{z}_{i,t} = (z_1, \ldots, z_{i-1}, t, z_i, \ldots, z_{d-1}) \in \mathbb{R}^d$. Note that for every $\mathbf{z} \in \mathbb{R}^{d-1}$ and every $t, t' \in \mathbb{R}$, we have $\mathbf{z}_{i,t} \in G_i$ iff $\mathbf{z}_{i,t'} \in G_i$. Let $G'_i = \{\mathbf{z} \in \mathbb{R}^{d-1} : \exists t \in \mathbb{R} \text{ s.t. } \mathbf{z}_{i,t} \in G_i\}$. Now, we have

$$
Pr(A_i \cap (\mathbb{R}^d \setminus G_i) \cap [-R, R]^d) = \int_{A_i \cap (\mathbb{R}^d \setminus G_i) \cap [-R,R]^d} \mu(\mathbf{x}) d\mathbf{x}
$$

$$
= \int_{\mathbb{R}^{d-1} \setminus G'_i} \left[ \int_{\{t \in \mathbb{R} : \mathbf{z}_{i,t} \in A_i \cap [-R,R]^d\}} \mu(\mathbf{z}_{i,t}) dt \right] d\mathbf{z}
$$

$$
= \int_{\mathbb{R}^{d-1} \setminus G'_i} \left[ \int_{\{t \in \mathbb{R} : \mathbf{z}_{i,t} \in A_i \cap [-R,R]^d\}} \mu_{[d] \setminus i}(\mathbf{z}) \mu_{i | [d] \setminus i}(t | \mathbf{z}) dt \right] d\mathbf{z}
$$

$$
= \int_{\mathbb{R}^{d-1} \setminus G'_i} \mu_{[d] \setminus i}(\mathbf{z}) \left[ \int_{\{t \in \mathbb{R} : \mathbf{z}_{i,t} \in A_i \cap [-R,R]^d\}} \mu_{i | [d] \setminus i}(t | \mathbf{z}) dt \right] d\mathbf{z}
$$

$$
\leq \sup_{\mathbf{z} \in \mathbb{R}^{d-1} \setminus G'_i} \int_{\{t \in \mathbb{R} : \mathbf{z}_{i,t} \in A_i \cap [-R,R]^d\}} \mu_{i | [d] \setminus i}(t | \mathbf{z}) dt \ .
$$

Recall that if $\mathbf{z} \in \mathbb{R}^{d-1} \setminus G'_i$ then $\mu_{i | [d] \setminus i}(t | \mathbf{z}) \leq M$ for all $t \in \mathbb{R}$. Hence the above is at most

$$
\sup_{\mathbf{z} \in \mathbb{R}^{d-1} \setminus G'_i} \int_{\{t \in \mathbb{R} : \mathbf{z}_{i,t} \in A_i \cap [-R,R]^d\}} M dt \ .
$$

Also, recall that if $\mathbf{x} \in A_i \cap [-R, R]^d$ then $x_i \in [a'_j, b'_j]$ for some $[a'_j, b'_j] \in I'_{i, \mathbf{x}^i, \gamma}$. Therefore the above is at most

$$
\sup_{\mathbf{z} \in \mathbb{R}^{d-1} \setminus G'_i} \int_{\{t \in [a'_j, b'_j] : [a'_j, b'_j] \in I'_{i, \mathbf{z}, \gamma}\}} M dt \leq \sup_{\mathbf{z} \in \mathbb{R}^{d-1} \setminus G'_i} \sum_{[a'_j, b'_j] \in I'_{i, \mathbf{z}, \gamma}} \int_{[a'_j, b'_j]} M dt
$$

$$
= \sup_{\mathbf{z} \in \mathbb{R}^{d-1} \setminus G'_i} \sum_{[a'_j, b'_j] \in I'_{i, \mathbf{z}, \gamma}} (b'_j - a'_j) M
$$

$$
= \sup_{\mathbf{z} \in \mathbb{R}^{d-1} \setminus G'_i} M \cdot \ell(I'_{i, \mathbf{z}, \gamma}) \overset{(Eq.\ 7)}{<} M \left( \frac{2l}{p(d)} + \frac{2Bl}{\gamma} \right) \ .
$$

Now, we have

$$
Pr\left(A_i \cap (\mathbb{R}^d \setminus G_i) | [-R, R]^d\right) = \frac{Pr\left(A_i \cap (\mathbb{R}^d \setminus G_i) \cap [-R, R]^d\right)}{Pr([-R, R]^d)}
$$

$$
\leq M \left( \frac{2l}{p(d)} + \frac{2Bl}{\gamma} \right) \frac{1}{1 - \delta} \ .
$$

Combining the above with Eq. 4 and 5, and using $\delta \leq \frac{1}{2}$, $\delta' = \frac{\epsilon^2}{1600 B^2 d}$ and $\gamma = \frac{6400 B^3 dMl}{\epsilon^2}$, we have

$$
Pr\left(A_i | [-R, R]^d\right) \leq \frac{\delta'}{1 - \delta} + M \left( \frac{2l}{p(d)} + \frac{2Bl}{\gamma} \right) \frac{1}{1 - \delta} \leq 2\delta' + 2M \left( \frac{2l}{p(d)} + \frac{2Bl}{\gamma} \right)
$$

$$
= \frac{\epsilon^2}{800 B^2 d} + \frac{4Ml}{p(d)} + \frac{\epsilon^2}{1600 B^2 d} \ .
$$

Therefore, for a sufficiently large polynomial $p(d)$ we have $Pr\left(A_i | [-R, R]^d\right) \leq \frac{\epsilon^2}{400 B^2 d}$. $\qquad\square$

By combining Lemma A.3 and Eq. 3, and plugging in $\delta = \frac{\epsilon^2}{400 B^2}$ we have

$$
Pr(A) \leq \sum_{i \in [d]} \frac{\epsilon^2}{400 B^2 d} + \frac{\epsilon^2}{400 B^2} = \frac{\epsilon^2}{200 B^2} \ .
$$

Finally, since for every $\mathbf{x}$ we have $(N'(\mathbf{x}) - \tilde{N}(\mathbf{x}))^2 \leq (2B)^2$, and since for every $\mathbf{x} \notin A$ we have $(N'(\mathbf{x}) - \tilde{N}(\mathbf{x}))^2 \leq \frac{\epsilon^2}{50}$, then

$$
\mathbb{E}_{\mathbf{x} \sim \mu} \left( N'(\mathbf{x}) - \tilde{N}(\mathbf{x}) \right)^2 \leq Pr(A) \cdot (2B)^2 + Pr(\mathbb{R}^d \setminus A) \cdot \frac{\epsilon^2}{50} \leq \frac{\epsilon^2}{200 B^2} \cdot 4B^2 + \frac{\epsilon^2}{50} = \left( \frac{\epsilon}{5} \right)^2 \ .
$$

### A.1.2 Proof of Lemma A.2

The network $\hat{N}$ consists of three parts. First, it transforms with high probability the input $\mathbf{x}$ to a binary representation of $\tilde{\mathbf{x}}$. Then, it simulates $N(\tilde{\mathbf{x}})$ by using arithmetic operations on binary vectors. Finally, it performs clipping of the output to the interval $[-B, B]$ and transforms it from the binary representation to its real value.

We start with the first part of $\hat{N}$, namely, transforming the input $\mathbf{x}$ to a binary representation of $\tilde{\mathbf{x}}$. The following lemma shows a property of almost-bounded conditional densities, that is required for this transformation.

**Lemma A.4.** *Let $\mu$ be a distribution with an almost-bounded conditional density. Then, for every $\epsilon = \frac{1}{\text{poly}(d)}$ there is $\Delta = \frac{1}{\text{poly}(d)}$ such that for every $i \in [d]$ and $s \in \mathbb{R}$ we have*

$$Pr_{\mathbf{x}\sim\mu}\left(x_i \in [s, s+\Delta]\right) \leq \epsilon \ .$$

*Proof.* For $\mathbf{x} \in \mathbb{R}^d$ we denote $\mathbf{x}^i = (x_1, \ldots, x_{i-1}, x_{i+1}, \ldots, x_d) \in \mathbb{R}^{d-1}$. Since $\mu$ has an almost-bounded conditional density, then there is $M = \text{poly}(d)$ such that for every $i \in [d]$ we have

$$Pr_{\mathbf{x}\sim\mu}\left(\exists t \in \mathbb{R} \text{ s.t. } \mu_{i|[d]\backslash i}(t|\mathbf{x}^i) > M\right) \leq \frac{\epsilon}{2} \ .$$

Let $\Delta = \frac{1}{\text{poly}(d)}$ such that $M\Delta \leq \frac{\epsilon}{2}$.

Then,

$$Pr_{\mathbf{x}\sim\mu}(x_i \in [s, s+\Delta]) = \int_{\{\mathbf{x}:x_i\in[s,s+\Delta]\}} \mu(\mathbf{x})d\mathbf{x}$$

$$= \int_{\{\mathbf{x}:x_i\in[s,s+\Delta]\}} \mu_{[d]\backslash i}(\mathbf{x}^i)\mu_{i|[d]\backslash i}(x_i|\mathbf{x}^i)d\mathbf{x}$$

$$= \int_{\mathbb{R}^{d-1}} \mu_{[d]\backslash i}(\mathbf{x}^i) \left[\int_{[s,s+\Delta]} \mu_{i|[d]\backslash i}(x_i|\mathbf{x}^i)dx_i\right] d\mathbf{x}^i$$

$$= \int_{\{\mathbf{x}^i\in\mathbb{R}^{d-1}:\forall t\in\mathbb{R} \ . \ \mu_{i|[d]\backslash i}(t|\mathbf{x}^i)\leq M\}} \mu_{[d]\backslash i}(\mathbf{x}^i) \left[\int_{[s,s+\Delta]} \mu_{i|[d]\backslash i}(x_i|\mathbf{x}^i)dx_i\right] d\mathbf{x}^i +$$

$$\int_{\{\mathbf{x}^i\in\mathbb{R}^{d-1}:\exists t\in\mathbb{R} \ . \ \mu_{i|[d]\backslash i}(t|\mathbf{x}^i)>M\}} \mu_{[d]\backslash i}(\mathbf{x}^i) \left[\int_{[s,s+\Delta]} \mu_{i|[d]\backslash i}(x_i|\mathbf{x}^i)dx_i\right] d\mathbf{x}^i$$

$$\leq \int_{\{\mathbf{x}^i\in\mathbb{R}^{d-1}:\forall t\in\mathbb{R} \ . \ \mu_{i|[d]\backslash i}(t|\mathbf{x}^i)\leq M\}} \mu_{[d]\backslash i}(\mathbf{x}^i) \left[\int_{[s,s+\Delta]} M dx_i\right] d\mathbf{x}^i +$$

$$\int_{\{\mathbf{x}^i\in\mathbb{R}^{d-1}:\exists t\in\mathbb{R} \ . \ \mu_{i|[d]\backslash i}(t|\mathbf{x}^i)>M\}} \mu_{[d]\backslash i}(\mathbf{x}^i) \left[\int_{\mathbb{R}} \mu_{i|[d]\backslash i}(x_i|\mathbf{x}^i)dx_i\right] d\mathbf{x}^i$$

$$\leq \int_{\{\mathbf{x}^i\in\mathbb{R}^{d-1}:\forall t\in\mathbb{R} \ . \ \mu_{i|[d]\backslash i}(t|\mathbf{x}^i)\leq M\}} \mu_{[d]\backslash i}(\mathbf{x}^i)M\Delta d\mathbf{x}^i +$$

$$\int_{\{\mathbf{x}\in\mathbb{R}^d:\exists t\in\mathbb{R} \ . \ \mu_{i|[d]\backslash i}(t|\mathbf{x}^i)>M\}} \mu(\mathbf{x})d\mathbf{x}$$

$$\leq M\Delta + Pr_{\mathbf{x}\sim\mu}\left(\exists t \in \mathbb{R} \text{ s.t. } \mu_{i|[d]\backslash i}(t|\mathbf{x}^i) > M\right) \leq \frac{\epsilon}{2} + \frac{\epsilon}{2} = \epsilon \ .$$

$\square$

Let $c$ be an integer greater or equal to $\log(2Rp(d) + 1)$. For $i \in [d]$ we denote by $(p(d)\tilde{x}_i)^{\text{bin}(c)} \in \{0,1\}^c$ the $c$-bits binary representation of the integer $p(d)\tilde{x}_i$. Note that since $p(d)\tilde{x}_i \in [-Rp(d), Rp(d)]$ then $c$ bits are sufficient. We use the standard *two's complement* binary representation. In this representation, the arithmetic operations of addition and multiplication of signed numbers are identical to those for unsigned numbers. Thus, we do not need to handle negative and positive numbers differently. We denote by $(p(d)\tilde{\mathbf{x}})^{\text{bin}(c)} \in \{0,1\}^{c\cdot d}$ the binary representation of $p(d)\tilde{\mathbf{x}}$, obtained by concatenating $(p(d)\tilde{x}_i)^{\text{bin}(c)}$ for $i = 1, \ldots, d$.

**Lemma A.5.** *Let $c \leq \mathrm{poly}(d)$ be an integer greater or equal to $\log(2Rp(d)+1)$ and let $\delta' = \frac{1}{\mathrm{poly}(d)}$. There is a neural network $\mathcal{N}$ of depth $2$, width $\mathrm{poly}(d)$, weights bounded by some $\mathrm{poly}(d)$, and $(c \cdot d)$ outputs, such that*

$$Pr_{\mathbf{x} \sim \mu}\left(\mathcal{N}(\mathbf{x}) = (p(d)\tilde{\mathbf{x}})^{\mathrm{bin}(c)}\right) \geq 1 - \delta' \;.$$

*Proof.* In order to construct $\mathcal{N}$, we need to show how to compute $(p(d)\tilde{x}_i)^{\mathrm{bin}(c)}$ for every $i \in [d]$. We will show a depth-2 network $\mathcal{N}'$ such that given $x_i \sim \mu_i$ it outputs $(p(d)\tilde{x}_i)^{\mathrm{bin}(c)}$ w.p. $\geq 1 - \frac{\delta'}{d}$. Then, the network $\mathcal{N}$ consists of $d$ copies of $\mathcal{N}'$, and satisfies

$$Pr_{\mathbf{x} \sim \mu}\left(\mathcal{N}(\mathbf{x}) \neq (p(d)\tilde{\mathbf{x}})^{\mathrm{bin}(c)}\right) \leq \sum_{i \in [d]} Pr_{x_i \sim \mu_i}\left(\mathcal{N}'(x_i) \neq (p(d)\tilde{x}_i)^{\mathrm{bin}(c)}\right) \leq \frac{\delta'}{d} \cdot d = \delta' \;.$$

For $j \in [c]$ let $I_j \subseteq \{-Rp(d), \ldots, Rp(d)\}$ be the integers such that the $j$-th bit in their binary representation is $1$. Hence, given $x_i$, the network $\mathcal{N}'$ should output in the $j$-th output $\mathbb{1}_{I_j}(p(d)\tilde{x}_i)$.

By Lemma A.4, there is $\Delta = \frac{1}{\mathrm{poly}(d)}$ such that for every $i \in [d]$ and every $t \in \mathbb{R}$ we have

$$Pr_{\mathbf{x} \sim \mu}\left(x_i \in \left[t - \frac{\Delta}{p(d)}, t + \frac{\Delta}{p(d)}\right]\right) \leq \frac{\delta'}{(2Rp(d)+2)d} \;. \tag{8}$$

For an integer $-Rp(d) \leq l \leq Rp(d)$, let $g_l : \mathbb{R} \to \mathbb{R}$ be such that

$$g_l(t) = \left[\frac{1}{\Delta}\left(t - l + \frac{1}{2}\right)\right]_+ - \left[\frac{1}{\Delta}\left(t - l + \frac{1}{2} - \Delta\right)\right]_+ \;.$$

Note that $g_l(t) = 0$ if $t \leq l - \frac{1}{2}$, and that $g_l(t) = 1$ if $t \geq l - \frac{1}{2} + \Delta$. Let $g'_l(t) = g_l(t) - g_{l+1}(t)$. Note that $g'_l(t) = 0$ if $t \leq l - \frac{1}{2}$ or $t \geq l + \frac{1}{2} + \Delta$, and that $g'_l(t) = 1$ if $l - \frac{1}{2} + \Delta \leq t \leq l + \frac{1}{2}$.

Let $h_j(t) = \sum_{l \in I_j} g'_l(t)$. Note that for every $l \in \{-Rp(d), \ldots, Rp(d)\}$ and $l - \frac{1}{2} + \Delta \leq t \leq l + \frac{1}{2}$ we have $h_j(t) = 1$ if $l \in I_j$ and $h_j(t) = 0$ otherwise. Hence, for $p(d)x_i \in [-Rp(d) - \frac{1}{2}, Rp(d) + \frac{1}{2}]$, if $|p(d)x_i - p(d)\tilde{x}_i| \leq \frac{1}{2} - \Delta$ then $h_j(p(d)x_i) = \mathbb{1}_{I_j}(p(d)\tilde{x}_i)$. For $p(d)x_i \leq -Rp(d) - \frac{1}{2} - \Delta$ and for $p(d)x_i \geq Rp(d) + \frac{1}{2} + \Delta$, we have $\tilde{x}_i = 0$ and $h_j(p(d)x_i) = 0 = \mathbb{1}_{I_j}(0) = \mathbb{1}_{I_j}(p(d)\tilde{x}_i)$. Therefore, if $h_j(p(d)x_i) \neq \mathbb{1}_{I_j}(p(d)\tilde{x}_i)$ then $p(d)x_i \in [l - \frac{1}{2} - \Delta, l - \frac{1}{2} + \Delta]$ for some integer $-Rp(d) \leq l \leq Rp(d) + 1$.

Let $\mathcal{N}'$ be such that $\mathcal{N}'(x_i) = (h_1(p(d)x_i), \ldots, h_c(p(d)x_i))$. Note that $\mathcal{N}'$ can be implemented by a depth-2 neural network.

Now,

$$Pr_{x_i \sim \mu_i}\left(\mathcal{N}'(x_i) \neq (p(d)\tilde{x}_i)^{\mathrm{bin}(c)}\right) = Pr_{x_i \sim \mu_i}\left(\exists j \in [c] \text{ s.t. } h_j(p(d)x_i) \neq (p(d)\tilde{x}_i)^{\mathrm{bin}(c)}_j\right)$$

$$= Pr_{x_i \sim \mu_i}\left(\exists j \in [c] \text{ s.t. } h_j(p(d)x_i) \neq \mathbb{1}_{I_j}(p(d)\tilde{x}_i)\right)$$

$$\leq Pr_{x_i \sim \mu_i}\left(p(d)x_i \in \left[l - \frac{1}{2} - \Delta, l - \frac{1}{2} + \Delta\right], -Rp(d) \leq l \leq Rp(d) + 1\right)$$

$$\leq \sum_{-Rp(d) \leq l \leq Rp(d)+1} Pr_{x_i \sim \mu_i}\left(x_i \in \left[\frac{l}{p(d)} - \frac{1}{2p(d)} - \frac{\Delta}{p(d)}, \frac{l}{p(d)} - \frac{1}{2p(d)} + \frac{\Delta}{p(d)}\right]\right)$$

$$\stackrel{(Eq.\ 8)}{\leq} (2Rp(d)+2) \cdot \frac{\delta'}{(2Rp(d)+2)d} = \frac{\delta'}{d} \;.$$

$\square$

We now show that $\tilde{\mathbf{x}} \mapsto N(\tilde{\mathbf{x}})$ for our network $N$ can be computed approximately by a depth-$k$ network $N''$ whose weights and biases are at most $2^{\mathrm{poly}(d)}$, and have a binary representation with $\mathrm{poly}(d)$ bits. The network $N''$ will be useful later in order to simulate such a computation with arithmetic operations on binary vectors.

**Lemma A.6.** *([18]) Consider a system $A\mathbf{x} \leq \mathbf{b}$ of arbitrary finite number of linear inequalities in $l$ variables. Assume that all entries in $A$ and $\mathbf{b}$ are integers of absolute value at most $a$. If this system has a solution in $\mathbb{R}^l$, then it has a solution of the form $\left(\frac{s_1}{t}, \ldots, \frac{s_l}{t}\right)$, where $s_1, \ldots, s_l, t$ are integers of absolute value at most $(2l + 1)!a^{2l+1}$.*

**Lemma A.7.** *Let $p'(d) = \mathrm{poly}(d)$. There is a $\mathrm{poly}(d)$-sized neural network $N''$ of depth $k$ such that for every $\tilde{\mathbf{x}} \in \mathcal{I}^d$ we have:*

- *If $N(\tilde{\mathbf{x}}) \in [-B, B]$ then $|N''(\tilde{\mathbf{x}}) - N(\tilde{\mathbf{x}})| \leq \frac{1}{p'(d)}$.*

- *If $N(\tilde{\mathbf{x}}) > B$ then $N''(\tilde{\mathbf{x}}) \geq B$.*

- *If $N(\tilde{\mathbf{x}}) < -B$ then $N''(\tilde{\mathbf{x}}) \leq -B$.*

*Moreover, $N''$ satisfies the following:*

- *There is a positive integer $t \leq 2^{\mathrm{poly}(d)}$ such that all weights and biases are in $Q_t = \{\frac{s}{t} : |s| \leq 2^{\mathrm{poly}(d)}, s \in \mathbb{Z}\}$.*

- *The weights in layers $2, \ldots, k$ are all in $\{-1, 1\}$.*

*Proof.* In [18] it is shown that a similar property holds for the case where the output neuron has sign activation, namely, where the output of the network is Boolean. We extend this result to real-valued functions.

We construct $N''$ in three steps. First, we transform $N$ into a network $N_1$ of depth $k$ where the fan-out of each hidden neuron is 1, such that for every $\mathbf{x} \in \mathbb{R}^d$ we have $N_1(\mathbf{x}) = N(\mathbf{x})$. Then, we transform $N_1$ into a network $N_2$ of depth $k$ where the weights in layers $2, \ldots, k$ are all in $\{-1, 1\}$, such that for every $\mathbf{x} \in \mathbb{R}^d$ we have $N_2(\mathbf{x}) = N_1(\mathbf{x})$. Finally, we show that $N_2$ can be transformed to a network $N''$ that satisfies the requirements (in particular, with exponentially-bounded weights and biases). The last stage is the most delicate one, and can be roughly described as follows: We create a huge set of linear inequalities, which encodes the requirement that the weights and biases of each neuron in $N_2$ produce the appropriate outputs, separately for each and every possible input $\tilde{\mathbf{x}}$ from our grid $\mathcal{I}^d$ (up to polynomially small error). Moreover, it can be shown that the size of the elements in our linear inequalities is $\mathrm{poly}(d)$. Hence, invoking Lemma A.6, we get that there is a solution to the linear system (namely, a set of weights and biases) which approximate $N_2$, yet have only $2^{\mathrm{poly}(d)}$-sized entries.

We now turn to the formal proof. First, the network $N_1$ is obtained by proceeding inductively from the output neuron towards the input neurons. Each hidden neuron with fan-out $c > 1$ is duplicated $c$ times. Let $l_i, l_i'$ be the number of hidden neurons in the $i$-th layer of $N$ and $N_1$ respectively. Note that $l_i' \leq l_i \cdot l_{i+1}'$. Since $k$ is constant and $l_i = \mathrm{poly}(d)$ then the size of $N_1$ is also $\mathrm{poly}(d)$.

In order to construct $N_2$, we, again, proceed inductively from the output neuron $n_{\mathrm{out}}$ of $N_1$ towards the input neurons. Let $w_1, \ldots, w_l$ be the weights of the output neuron and let $n_1, \ldots, n_l$ be the corresponding hidden neurons. That is, for each $i \in [l]$ there is an edge with weight $w_i \neq 0$ between $n_i$ and $n_{\mathrm{out}}$. Now, for each $i \in [l]$, we replace the weight $w_i$ of the edge $(n_i, n_{\mathrm{out}})$ by $\frac{w_i}{|w_i|}$, and multiply the weights and bias of $n_i$ by $|w_i|$. Note that now the multiplication by $|w_i|$ is done before $n_i$ instead of after it, but $n_{\mathrm{out}}$ still receives the same input as in $N_1$. Since the fan-out of every hidden neuron in $N_1$ is 1, we can repeat the same operation also in the predecessors of $n_1, \ldots, n_l$, and continue until the first hidden layer. Hence, we obtain a network $N_2$ where the weights in layers $2, \ldots, k$ are all in $\{-1, 1\}$.

We now show that $N_2$ can be transformed to a network $N''$ that satisfies the requirements. Let $l_1$ be the number of neurons in the first hidden layer of $N_2$, let $m_w = d \cdot l_1$ be the number of weights in the first layer (including 0 weights), and let $m_b$ the number of hidden neurons in $N_2$, that is, the number of biases in $N_2$. Let $m = m_w + m_b$. For each $i \in [l_1]$ we denote by $\mathbf{w}_i \in \mathbb{R}^d$ the weights of the $i$-th neuron in the first hidden layer in $N_2$, and for each hidden neuron $n$ in $N_2$ we denote by $b_n$ the bias of $n$. We define a linear system $A\mathbf{z} \leq \mathbf{c}$ where the variables $\mathbf{z} \in \mathbb{R}^m$ correspond to the weights of the first layer and the biases in $N_2$. We denote by $\mathbf{z}_i$ the $d$ variables in $\mathbf{z}$ that correspond to $\mathbf{w}_i$, and by $z_n$ the variable in $\mathbf{z}$ that corresponds to $b_n$. Note that each assignment to the variables $\mathbf{z}$

induces a neural network $N_2^{\mathbf{z}}$ where the weights in the first layer and the biases in $N_2$ are replaced by the corresponding variables.

For each $\tilde{\mathbf{x}} \in \mathcal{I}^d$ we place in the system $A\mathbf{z} \leq \mathbf{c}$ an inequality for each hidden neuron in $N_2$, and either one or two inequalities for the output neuron. These inequalities are defined by induction on the depth of the neuron. If $n_i$ is the $i$-th neuron in the first hidden layer and its input in the computation of $N_2(\tilde{\mathbf{x}})$ satisfies $\langle \tilde{\mathbf{x}}, \mathbf{w}_i \rangle + b_{n_i} \geq 0$, then we add the inequality $\langle \tilde{\mathbf{x}}, \mathbf{z}_i \rangle + z_{n_i} \geq 0$ to the system. Otherwise, we add the inequality $\langle \tilde{\mathbf{x}}, \mathbf{z}_i \rangle + z_{n_i} \leq 0$. Note that the variables in the inequality are $\mathbf{z}_i, z_{n_i}$, and that $\tilde{\mathbf{x}}$ is a constant. Let $S_1 \subseteq \{n_i : i \in [l_1]\}$ be the neurons in the first hidden layer where $\langle \tilde{\mathbf{x}}, \mathbf{w}_i \rangle + b_{n_i} \geq 0$, that is, the neurons where the ReLU is active in the computation $N_2(\tilde{\mathbf{x}})$. Now, the input for each neuron $n'$ in the second hidden layer in the computation $N_2(\tilde{\mathbf{x}})$, is of the form $I(n') = \sum_{n_i \in S_1} a_i(\langle \tilde{\mathbf{x}}, \mathbf{w}_i \rangle + b_{n_i}) + b_{n'}$ where $a_i \in \{-1, 0, 1\}$ is the weight of the edge $(n_i, n')$ in $N_2$. Let $I'(n') = \sum_{n_i \in S_1} a_i(\langle \tilde{\mathbf{x}}, \mathbf{z}_i \rangle + z_{n_i}) + z_{n'}$. If $I(n') \geq 0$ then we add the inequality $I'(n') \geq 0$, and otherwise we add $I'(n') \leq 0$. Note that the variables in the inequality are $\mathbf{z}_i, z_{n_i}, z_{n'}$ (for the appropriate indices $i$) and that $\tilde{\mathbf{x}}, a_i$ are constants. Thus, this inequality is linear.

We denote by $S_2$ the set of neurons in the second hidden layer where the ReLU is active in the computation $N_2(\tilde{\mathbf{x}})$, and for each neuron $n''$ in the third hidden layer we define $I(n'')$ and $I'(n'')$ and add a linear inequality analogously. We continue until we reach the output neuron $n_{\text{out}}$. Let $I(n_{\text{out}})$ be the input to $n_{\text{out}}$ in the computation $N_2(\tilde{\mathbf{x}})$, and let $I'(n_{\text{out}})$ be the corresponding linear expression, where the variables are $\mathbf{z}$ and the constants are $\tilde{\mathbf{x}}$ and the weights in layers $2, \ldots, k$ (which are all in $\{-1, 0, 1\}$). Note that $I(n_{\text{out}}) = N_2(\tilde{\mathbf{x}}) = N(\tilde{\mathbf{x}})$. If $N(\tilde{\mathbf{x}}) \in [-B, B]$, then let $-Bp'(d) \leq j \leq Bp'(d) - 1$ be an integer such that $\frac{j}{p'(d)} \leq I(n_{\text{out}}) \leq \frac{j+1}{p'(d)}$. Now, we add the two inequalities $\frac{j}{p'(d)} \leq I'(n_{\text{out}}) \leq \frac{j+1}{p'(d)}$, where $j, p'(d)$ are constants. If $N(\tilde{\mathbf{x}}) > B$, then we add the inequality $I'(n_{\text{out}}) \geq B$, and if $N(\tilde{\mathbf{x}}) < -B$, then we add the inequality $I'(n_{\text{out}}) \leq -B$.

Note that if $\mathbf{z}$ satisfies all the inequalities $A\mathbf{z} \leq \mathbf{c}$, then for each neuron $n$, the expression $I'(n)$ is consistent with the set of active ReLUs according to the inequalities of the previous layers. Therefore, the input to $n$ in the computation $N_2^{\mathbf{z}}(\tilde{\mathbf{x}})$ is $I'(n)$. Hence, for such $\mathbf{z}$ we have for every $\tilde{\mathbf{x}} \in \mathcal{I}^d$ that if $N_2(\tilde{\mathbf{x}}) \in [-B, B]$ then $|N_2^{\mathbf{z}}(\tilde{\mathbf{x}}) - N_2(\tilde{\mathbf{x}})| \leq \frac{1}{p'(d)}$, if $N_2(\tilde{\mathbf{x}}) > B$ then $N_2^{\mathbf{z}}(\tilde{\mathbf{x}}) \geq B$, and if $N_2(\tilde{\mathbf{x}}) < -B$ then $N_2^{\mathbf{z}}(\tilde{\mathbf{x}}) \leq -B$. Note that $A\mathbf{z} \leq \mathbf{c}$ has a solution in $\mathbb{R}^m$, since the weights and biases in $N_2$ satisfy all the inequalities. The entries in $A, \mathbf{c}$ are either integers with absolute value at most $\text{poly}(d)$, or of the form $q \cdot \tilde{x}_i = \frac{q'}{p(d)}$ or $\frac{q}{p'(d)}$ where $q, q'$ are integers with absolute values at most $\text{poly}(d)$. Therefore, by Lemma A.6, there is an integer $a = \text{poly}(d)$ such that the linear system $(p(d)p'(d)A)\mathbf{z} \leq p(d)p'(d)\mathbf{c}$ has a solution $\mathbf{z} = \left(\frac{s_1}{t}, \ldots, \frac{s_m}{t}\right)$, where $s_1, \ldots, s_m, t$ are integers of absolute value at most $(2m+1)! a^{2m+1} \leq 2^{\text{poly}(d)}$. Hence, the network $N'' = N_2^{\mathbf{z}}$ satisfies the requirements. $\qquad \square$

Let $N''$ be the network from Lemma A.7 with $p'(d) = \frac{\sqrt{50}}{\epsilon}$. The following lemma follows easily.

**Lemma A.8.** *For every* $\tilde{\mathbf{x}} \in \mathcal{I}^d$ *we have*

$$\left|[N''(\tilde{\mathbf{x}})]_{[-B,B]} - N'(\tilde{\mathbf{x}})\right| \leq \frac{1}{p'(d)} .$$

*Proof.*  • If $N(\tilde{\mathbf{x}}) \in [-B, B]$ then $|N''(\tilde{\mathbf{x}}) - N(\tilde{\mathbf{x}})| \leq \frac{1}{p'(d)}$ and we have

$$\left|[N''(\tilde{\mathbf{x}})]_{[-B,B]} - N'(\tilde{\mathbf{x}})\right| \leq |N''(\tilde{\mathbf{x}}) - N'(\tilde{\mathbf{x}})| = |N''(\tilde{\mathbf{x}}) - N(\tilde{\mathbf{x}})| \leq \frac{1}{p'(d)} .$$

• If $N(\tilde{\mathbf{x}}) > B$ then $N''(\tilde{\mathbf{x}}) \geq B$, and therefore

$$\left|[N''(\tilde{\mathbf{x}})]_{[-B,B]} - N'(\tilde{\mathbf{x}})\right| = |B - B| = 0 .$$

• If $N(\tilde{\mathbf{x}}) < -B$ then $N''(\tilde{\mathbf{x}}) \leq -B$, and therefore

$$\left|[N''(\tilde{\mathbf{x}})]_{[-B,B]} - N'(\tilde{\mathbf{x}})\right| = |-B - (-B)| = 0 .$$

$\qquad \square$

The weights and biases in $N''$ might be exponential, but they have a binary representation with $\mathrm{poly}(d)$ bits. This property enables us to simulate $[N''(\tilde{\mathbf{x}})]_{[-B,B]}$ using arithmetic operations on binary vectors.

We now show how to simulate $[N''(\tilde{\mathbf{x}})]_{[-B,B]}$ using binary operations. Recall that the input $\tilde{\mathbf{x}}$ to $N''$ is such that every component $\tilde{x}_i$ is of the form $\frac{q_i}{p(d)}$ for some integer $q_i$ with absolute value at most $\mathrm{poly}(d)$. We will represent each component in the input by the binary representation of the integer $p(d)\tilde{x}_i$. It implies that while simulating $N''$, we should replace each weight $w$ in the first layer of $N''$ with $w' = \frac{w}{p(d)}$. Then, $w \cdot \tilde{x}_i = w' \cdot (p(d)\tilde{x}_i)$. Recall that the network $N''$ is such that all weights in layers $2, \ldots, k$ in $N''$ are in $\{-1, 1\}$ and all weights in the first layer and biases are of the form $\frac{s_i}{t}$ for some positive integer $t \leq 2^{\mathrm{poly}(d)}$, and integers $s_i$ with $|s_i| \leq 2^{\mathrm{poly}(d)}$. We represent each number of the form $\frac{v}{t}$ by the binary representation of $v$. Since for all weights and biases $\frac{s_i}{t}$ in $N''$ we can multiply both $t$ and $s_i$ by $p(d)$, we can assume w.l.o.g. that $p(d) \mid s_i$ and $p(d) \mid t$. Then, for each weight $w = \frac{s_i}{t}$ in the first layer of $N''$, we represent $w' = \frac{w}{p(d)} = \frac{s_i}{t \cdot p(d)}$ by the binary representation of the integer $\frac{s_i}{p(d)}$.

Since the input to a neuron in the first hidden layer of $N''$ is a sum of the form $I = \sum_{i \in [d]} w_i \tilde{x}_i + b = \sum_{i \in [d]} w_i'(p(d)\tilde{x}_i) + b$, then in order to simulate it we need to compute multiplications and additions of binary vectors. Note that $p(d)\tilde{x}_i$ are integers, $w_i'$ are represented by the binary representation of the integers $q_i$ such that $w_i' = \frac{q_i}{t}$, and $b$ is represented by the binary representation of the integer $q$ such that $b = \frac{q}{t}$. Then, $I$ is also of the form $\frac{v}{t}$ for an integer $v$ with $|v| \leq 2^{\mathrm{poly}(d)}$, and therefore it can be represented by the binary representation of $v$. Since the biases in $N''$ are of the form $\frac{s_i}{t}$ for integers $s_i$, and the weights in layers $2, \ldots, k$ are in $\{-1, 1\}$, then in the computation $N''(\tilde{\mathbf{x}})$ all values, namely, inputs to neurons in all layers, are of the form $\frac{v}{t}$ where $v$ is an integer with $|v| \leq 2^{\mathrm{poly}(d)}$. That is, a binary representation of $v$ requires $\mathrm{poly}(d)$ bits. Thus, since all values have $t$ in the denominator, then we ignore it and work only with the numerator.

Let $C' = \mathrm{poly}(d)$ be such that for all $\tilde{\mathbf{x}} \in \mathcal{I}^d$, all inputs to neurons in the computation $N''(\tilde{\mathbf{x}})$ are of the form $\frac{v}{t}$ where $v$ is an integer with absolute value at most $2^{C'}$. Namely, all values in the computation can be represented by $C'$ bits. Let $C = \mathrm{poly}(d)$ be such that every integer $v$ of absolute value at most $2^{C'} + Bt$ has a binary representation with $C$ bits. Also, assume that $C \geq \log(2Rp(d) + 1)$. Such $C$ will be sufficiently large in order to represent all inputs $p(d)\tilde{x}_i$ and all values in our simulation of $[N''(\tilde{\mathbf{x}})]_{[-B,B]}$.

We now show how to simulate $p(d)\tilde{\mathbf{x}} \mapsto [N''(\tilde{\mathbf{x}})]_{[-B,B]} + B$ with a threshold circuit.

**Lemma A.9.** *There is a threshold circuit $T$ of depth $3k + 1$, width $\mathrm{poly}(d)$, and $\mathrm{poly}(d)$-bounded weights, whose inputs are the $C$-bits binary representations of:*

- *$p(d)\tilde{x}_i$ for every $i \in [d]$.*

- *$\frac{s_i}{p(d)}$ and $\frac{-s_i}{p(d)}$ for every weight $\frac{s_i}{t}$ in the first layer of $N''$.*

*And its outputs are:*

- *The $C$-bits binary representation of $v$ such that:*

  - *If $N''(\tilde{\mathbf{x}}) \in [-B, B]$ then $\frac{v}{t} = N''(\tilde{\mathbf{x}}) + B$.*
  - *Otherwise $v = 0$.*

- *A bit $c$ such that $c = 1$ iff $N''(\tilde{\mathbf{x}}) > B$.*

*Proof.* In order to simulate the first layer of $N''$, we first need to compute a sum of the form $\sum_{i \in [d]} w_i \cdot z_i$ where $w_i = \frac{s_i}{p(d)}$ and $z_i = p(d)\tilde{x}_i$ are the inputs and are given in a binary representation. Hence, we are required to perform binary multiplications and then binary iterated addition, namely, addition of multiple numbers that are given by binary vectors. Binary iterated addition can be done by a depth-2 threshold circuit with polynomially-bounded weights and polynomial width, and binary multiplication can be done by a depth-3 threshold circuit with polynomially-bounded weights and polynomial width ([28]). The depth-3 circuit for multiplication shown in [28] first computes the partial products and then uses the depth-2 threshold circuit for iterated addition in order to compute

their sum. They show it for a multiplication of two $n$-bit numbers that results in a $2n$-bit number. The same method can be used also in our case for a multiplication of two $C$-bit numbers that results in a $C$-bit number, since $C$ was chosen such that we are guaranteed that there is no overflow. Also, in two's complement representation, multiplication and addition of signed numbers can be done similarly to the unsigned case. In our case, we need to compute multiplication and then iterated addition. Hence, instead of using a depth-5 threshold circuit that computes multiplications and then computes the iterated addition, we can use a depth-3 threshold circuit that first computes all partial products for all multiplications, and then computes a single iterated addition.

Since the hidden neurons in $N''$ have biases, we need to simulate sums of the form $b + \sum_{i \in [d]} w_i \cdot z_i$. Hence, the binary iterated addition should also include $b$. Therefore, the bias $b$ is hardwired into the circuit $T$. That is, for every bias $b = \frac{v}{t}$, we add $C$ gates to the first hidden layer with fan-in 0 and with biases in $\{0, 1\}$ that correspond to the binary representation of $v$.

Simulating the ReLUs of the first hidden layer in $N''$ can be done as follows. Let $v$ be an integer and let $v^{\text{bin}(C)} \in \{0, 1\}^C$ be its binary representation. Recall that in the two's complement representation the most significant bit (MSB) is 1 iff the number is negative. Now, we reduce the value of the MSB, namely $v_C^{\text{bin}(C)}$, from all other $C - 1$ bits $v_1^{\text{bin}(C)}, \ldots, v_{C-1}^{\text{bin}(C)}$. Thus, we transform $v^{\text{bin}(C)}$ to $\left( \text{sign}(v_1^{\text{bin}(C)} - v_C^{\text{bin}(C)}), \ldots, \text{sign}(v_{C-1}^{\text{bin}(C)} - v_C^{\text{bin}(C)}), 0 \right)$. Now, if $v < 0$, that is $v_C^{\text{bin}(C)} = 1$, then we obtain a binary vector whose bits are all 0. If $v \geq 0$ then $v_C^{\text{bin}(C)} = 0$ and therefore $v^{\text{bin}(C)}$ is not changed. Thus, simulating a ReLU of $N''$ requires one additional layer in the threshold circuit. Overall, the output of the first hidden layer of $N''$ can be computed by a depth-4 threshold circuit.

Now, the weights in layers $2, \ldots, k$ in $N''$ are in $\{-1, 1\}$. Note that simulating multiplication by a threshold circuit, as discussed above, requires 3 layers. However, we need to compute values of the form $b + \sum_i a_i \cdot z_i$ where $a_i \in \{-1, 1\}$, and $z_i, b$ are given by binary vectors. In order to avoid multiplication, we keep both the values of the computation $N''(\tilde{\mathbf{x}})$ in each layer, and their negations. That is, the circuit $T$ keeps both the binary representation of $z_i$ and the binary representation of $-z_i$, and then simulating each layer can be done by iterated addition, without binary multiplication. Keeping both $z_i$ and $-z_i$ in each layer is done as follows. When $T$ simulates the first layer of $N''$, it computes values of the form $z = b + \sum_{i \in [d]} w_i \cdot (p(d)\tilde{x}_i)$, and in parallel it should also compute $-z = -b + \sum_{i \in [d]} (-w_i) \cdot (p(d)\tilde{x}_i)$. Note that both $w_i$ and $-w_i$ are given as inputs to $T$, and that the binary representation of $v$ such that $-b = \frac{v}{t}$ can be hardwired into $T$, similarly to the case of $b$. Then, when simulating a ReLU of $N''$, it reduces the MSB of $z$ also from all bits of the binary representation of $-z$. Thus, if $z < 0$ then both $z$ and $-z$ become 0. Now, computing $z' = b' + \sum_i a_i \cdot z_i$ where $a_i \in \{-1, 1\}$ and $z_i, b'$ are binary numbers, can be done by iterated addition, and also computing $-z' = -b' + \sum_i -a_i \cdot z_i$ can be done by iterated addition. Note that the binary representations of $\pm v$ such that $b' = \frac{v}{t}$ are also hardwired into $T$. Since iterated addition can be implemented by a threshold circuit of depth 2, the sum $z' = b' + \sum_i a_i \cdot z_i$ can be implemented by 2 layers in $T$, and then implementing $[z']_+$, requires one more layer as discussed above. Thus, each of the layers $2, \ldots, k - 1$ in $N''$ requires 3 layers in $T$.

Let $N''(\tilde{\mathbf{x}}) = \frac{v}{t}$. When simulating the final layer of $N''$, we also add (as a part of the iterated addition) the hardwired binary representation of $Bt$. That is, instead of computing the binary representation of $v$, we compute the binary representation of $v' = v + Bt$. We also compute the binary representation of $v'' = -v + Bt$. Note that $\frac{v'}{t} = N''(\tilde{\mathbf{x}}) + B$ and $\frac{v''}{t} = -N''(\tilde{\mathbf{x}}) + B$. Now, the bit $c$ that $T$ should output is the MSB of $v''$, since $v''$ is negative iff $N''(\tilde{\mathbf{x}}) > B$. The $C$-bits binary vector that $T$ outputs is obtained from $v', v''$ by adding one final layer as follows. Let $\text{MSB}(v')$ and $\text{MSB}(v'')$ be the MSBs of $v', v''$. In the final layer we reduce $\text{MSB}(v') + \text{MSB}(v'')$ from all bits of $v'$. That is, if either $v'$ or $v''$ are negative, then we output 0, and otherwise we output $v'$. Now, if $N''(\tilde{\mathbf{x}}) \in [-B, B]$ then $v', v'' \in [0, 2Bt]$, and we output $v'$, which corresponds to $N''(\tilde{\mathbf{x}}) + B$. If $N''(\tilde{\mathbf{x}}) < -B$ then $\frac{v'}{t} = N''(\tilde{\mathbf{x}}) + B < 0$, and therefore $\text{MSB}(v') = 1$, and we output 0. If $N''(\tilde{\mathbf{x}}) > B$ then $\frac{v''}{t} = -N''(\tilde{\mathbf{x}}) + B < 0$, and therefore $\text{MSB}(v'') = 1$, and we output 0. Thus, simulating the final layer of $N''$ requires 3 layers in $T$: 2 layers for the iterated addition, and one layer for transforming $v', v''$ to the required output.

Finally, the depth of $T$ is $3k + 1$ since simulating the first layer of $N''$ requires 4 layers in $T$, and each additional layer in $N''$ required 3 layers in $T$. $\qquad \square$

The following simple lemma shows that threshold circuits can be transformed to neural networks.

**Lemma A.10.** *Let $T$ be a threshold circuit with $d$ inputs, $q$ outputs, depth $m$ and width $\mathrm{poly}(d)$. There is a neural network $\mathcal{N}$ with $q$ outputs, depth $m+1$ and width $\mathrm{poly}(d)$, such that for every $\mathbf{x} \in \{0,1\}^d$ we have $\mathcal{N}(\mathbf{x}) = T(\mathbf{x})$. If $T$ has $\mathrm{poly}(d)$-bounded weights then $\mathcal{N}$ also has $\mathrm{poly}(d)$-bounded weights. Moreover, for every input $\mathbf{x} \in \mathbb{R}^d$ the outputs of $\mathcal{N}$ are in $[0,1]$.*

*Proof.* Let $g$ be a gate in $T$, and let $\mathbf{w} \in \mathbb{Z}^l$ and $b \in \mathbb{Z}$ be its weights and bias. Let $n_1$ be a neuron with weights $\mathbf{w}$ and bias $b$, and let $n_2$ be a neuron with weights $\mathbf{w}$ and bias $b-1$. Let $\mathbf{y} \in \{0,1\}^l$. Since $(\langle \mathbf{w}, \mathbf{y} \rangle + b) \in \mathbb{Z}$, we have $[\langle \mathbf{w}, \mathbf{y} \rangle + b]_+ - [\langle \mathbf{w}, \mathbf{y} \rangle + b - 1]_+ = \mathrm{sign}(\langle \mathbf{w}, \mathbf{y} \rangle + b)$. Hence, the gate $g$ can be replaced by the neurons $n_1, n_2$. We replace all gates in $T$ by neurons and obtain a network $\mathcal{N}$. Since each output gate of $T$ is also replaced by two neurons, $\mathcal{N}$ has $m+1$ layers. Since for every $\mathbf{x} \in \mathbb{R}^d$, weight vector $\mathbf{w}$ and bias $b$ we have $[\langle \mathbf{w}, \mathbf{x} \rangle + b]_+ - [\langle \mathbf{w}, \mathbf{x} \rangle + b - 1]_+ \in [0,1]$ then for every input $\mathbf{x} \in \mathbb{R}^d$ the outputs of $\mathcal{N}(\mathbf{x})$ are in $[0,1]$. □

We are now ready to construct the network $\hat{N}$. Let $\delta' = \frac{\epsilon^2}{50 \cdot 36 B^2}$. The network $\hat{N}$ is such that w.p. at least $1 - \delta'$ we have $\hat{N}(\mathbf{x}) = [N''(\tilde{\mathbf{x}})]_{[-B,B]}$. It consists of three parts.

First, $\hat{N}$ transforms w.p. $\geq 1 - \delta'$ the input $\mathbf{x}$ to the $(C \cdot d)$-bits binary representation of $p(d)\tilde{\mathbf{x}}$. By Lemma A.5, it can be done with a 2-layers neural network $\mathcal{N}_1$.

Second, let $T$ be the threshold circuit from Lemma A.9. By Lemma A.10, $T$ can be implemented by a neural network $\mathcal{N}_2$ of depth $3k+2$. Note that the input to $\mathcal{N}_2$ has two parts:

1. The $(C \cdot d)$-bits binary representation of $p(d)\tilde{\mathbf{x}}$. This is the output of $\mathcal{N}_1$.

2. The binary representations of $\frac{\pm s_i}{p(d)}$ for every weight $\frac{s_i}{t}$ in the first layer of $N''$. This is hardwired into $\hat{N}$ by hidden neurons with fan-in 0 and appropriate biases in $\{0,1\}$.

Thus, using $\mathcal{N}_2$ the network $\hat{N}$ transforms the binary representation of $p(d)\tilde{\mathbf{x}}$ to the output $(v^{\mathrm{bin}(C)}, c)$ of $T$.

Third, $\hat{N}$ transforms $(v^{\mathrm{bin}(C)}, c)$ to $[N''(\tilde{\mathbf{x}})]_{[-B,B]}$ as follows. Let $v$ be the integer that corresponds to the binary vector $v^{\mathrm{bin}(C)}$. The properties of $v$ and $c$ from Lemma A.9, imply that $[N''(\tilde{\mathbf{x}})]_{[-B,B]} = \frac{v}{t} + c \cdot 2B - B$, since we have:

- If $N''(\tilde{\mathbf{x}}) \in [-B, B]$ then $\frac{v}{t} = N''(\tilde{\mathbf{x}}) + B$ and $c = 0$.
- If $N''(\tilde{\mathbf{x}}) < -B$ then $v = 0$ and $c = 0$.
- If $N''(\tilde{\mathbf{x}}) > B$ then $v = 0$ and $c = 1$.

Hence, we need to transform $(v^{\mathrm{bin}(C)}, c)$ to the real number $\frac{v}{t} + c \cdot 2B - B$. Note that $v \geq 0$, and therefore we have

$$\frac{v}{t} = \sum_{i \in [C-1]} v_i^{\mathrm{bin}(C)} \cdot \frac{2^{i-1}}{t} \ .$$

Also, note that $\frac{v}{t} \in [0, 2B]$, and therefore for every $i \in [C-1]$ we have $v_i^{\mathrm{bin}(C)} \cdot \frac{2^{i-1}}{t} \leq 2B$. Hence, we can ignore every $i > \log(2Bt) + 1$. Thus,

$$\frac{v}{t} = \sum_{i \in [\log(2Bt)+1]} v_i^{\mathrm{bin}(C)} \cdot \frac{2^{i-1}}{t} \ . \tag{9}$$

Since for $i \in [\log(2Bt) + 1]$ we have $\frac{2^{i-1}}{t} \leq 2B$, then in the above computation of $\frac{v}{t}$ the weights are positive numbers smaller or equal to $2B$. Thus, we can transform $(v^{\mathrm{bin}(C)}, c)$ to $\frac{v}{t} + c \cdot 2B - B$ in one layer with $\mathrm{poly}(d)$-bounded weights. In order to avoid bias in the output neuron, the additive term $-B$ is hardwired into $\hat{N}$ by adding a hidden neuron with fan-in 0 and bias 1 that is connected to the output neuron with weight $-B$.

Since the final layers of $\mathcal{N}_1$ and $\mathcal{N}_2$ do not have activations and can be combined with the next layers, and since the third part of $\hat{N}$ is a sum, then the depth of $\hat{N}$ is $3k+3$.

Thus, we have w.p. at least $1 - \delta'$ that $\hat{N}(\mathbf{x}) = [N''(\tilde{\mathbf{x}})]_{[-B,B]}$. By Lemma A.8, it implies that w.p. at least $1 - \delta'$ we have

$$\left| \hat{N}(\mathbf{x}) - \tilde{N}(\mathbf{x}) \right| \leq \frac{1}{p'(d)} . \tag{10}$$

However, it is possible (w.p. at most $\delta'$) that $\mathcal{N}_1$ fails to transform $\mathbf{x}$ to the binary representation of $p(d)\tilde{\mathbf{x}}$, and therefore the above inequality does not hold. Still, even in this case we can bound the output of $\hat{N}$ as follows. If $\mathcal{N}_1$ fails to transform $\mathbf{x}$ to $p(d)\tilde{\mathbf{x}}$, then the input to $\mathcal{N}_2$ may contains values other than $\{0, 1\}$. However, by Lemma A.10, the network $\mathcal{N}_2$ outputs $c$ and $v^{\mathrm{bin}(C)}$ such that each component is in $[0, 1]$. Now, when transforming $(v^{\mathrm{bin}(C)}, c)$ to $\frac{v}{t} + c \cdot 2B - B$ in the final layer of $\hat{N}$, we compute $\frac{v}{t}$ by the sum in Eq. 9. Since $v_i^{\mathrm{bin}(C)} \in [0, 1]$ for every $i$, this sum is at least 0 and at most

$$\frac{1}{t} \cdot 2^{\log(2Bt)+1} = 4B .$$

Therefore, the output of $\hat{N}$ is at most $4B + 1 \cdot 2B - B = 5B$, and at least $0 + 0 \cdot 2B - B = -B$. Thus, for every $\mathbf{x}$ we have $\hat{N}(\mathbf{x}) \in [-B, 5B]$. Since for every $\mathbf{x}$ we have $\tilde{N}(\mathbf{x}) \in [-B, B]$, then we have

$$\left| \hat{N}(\mathbf{x}) - \tilde{N}(\mathbf{x}) \right| \leq 6B .$$

Combining the above with Eq. 10 and plugging in $\delta' = \frac{\epsilon^2}{50 \cdot 36B^2}$ and $p'(d) = \frac{\sqrt{50}}{\epsilon}$, we have

$$\mathop{\mathbb{E}}_{\mathbf{x} \sim \mu} (\hat{N}(\mathbf{x}) - \tilde{N}(\mathbf{x}))^2 \leq (1 - \delta') \left( \frac{1}{p'(d)} \right)^2 + \delta' \cdot (6B)^2 = (1 - \delta') \frac{\epsilon^2}{50} + \frac{\epsilon^2}{50 \cdot 36B^2} \cdot 36B^2 \leq \left( \frac{\epsilon}{5} \right)^2 .$$

Therefore $\| \hat{N} - \tilde{N} \|_{L_2(\mu)} \leq \frac{\epsilon}{5}$ as required.

## A.2 Proof of Theorem 3.2

The proof follows the same ideas as the proof of Theorem 3.1, but is simpler. Consider the functions $f'$ and $N'$ that are defined in the proof of Theorem 3.1. For every $\mathbf{x} \in \mathcal{I}^d$ we denote $\tilde{\mathbf{x}} = \mathbf{x}$, and $\tilde{N}(\mathbf{x}) = N'(\tilde{\mathbf{x}}) = N'(\mathbf{x})$. Now, from the same arguments as in the proof of Theorem 3.1, it follows that we can bound $\| f' - f \|_{L_2(\mathcal{D})}$ and $\| N' - f' \|_{L_2(\mathcal{D})}$. Since $\| \tilde{N} - N' \|_{L_2(\mathcal{D})} = 0$, it remains to show that Lemma A.2 holds also in this case.

The network $\hat{N}$ will have a similar structure to the one in the proof of Lemma A.2.

First, it transforms the input $\mathbf{x}$ to the binary representation of $p(d)\tilde{\mathbf{x}} = p(d)\mathbf{x}$. This transformation is similar to the one from the proof of Lemma A.5. However, since $\mathcal{D}$ is such that for every $i \in [d]$ the component $x_i$ is of the form $\frac{j}{p(d)}$ for some integer $j$, then for an appropriate $\Delta$, we have for every integer $l$ that

$$Pr_{\mathbf{x} \sim \mathcal{D}} \left( x_i \in \left[ \frac{l}{p(d)} - \frac{1}{2p(d)} - \frac{\Delta}{p(d)}, \frac{l}{p(d)} - \frac{1}{2p(d)} + \frac{\Delta}{p(d)} \right] \right) = 0 .$$

Hence, there is a depth-2 network with $\mathrm{poly}(d)$ width and $\mathrm{poly}(d)$-bounded weights, that transforms $\mathbf{x}$ to the binary representation of $p(d)\tilde{\mathbf{x}}$ and succeeds w.p. 1.

Recall that in the proof of Lemma A.2, the next parts of $\hat{N}$ transform for every $\tilde{\mathbf{x}}$ the binary representation of $p(d)\tilde{\mathbf{x}}$ to $[N''(\tilde{\mathbf{x}})]_{[-B,B]}$. Since this transformation is already discrete and does not depend on the input distribution, we can also use it here. Then, by lemma A.8 we have for every $\tilde{\mathbf{x}}$ that

$$\left| [N''(\tilde{\mathbf{x}})]_{[-B,B]} - N'(\tilde{\mathbf{x}}) \right| \leq \frac{1}{p'(d)} .$$

Thus, for $p'(d) = \frac{5}{\epsilon}$, we obtain a network $\hat{N}$ such that w.p. 1 we have $\left| \hat{N}(\mathbf{x}) - \tilde{N}(\mathbf{x}) \right| \leq \frac{\epsilon}{5}$, and therefore $\| \hat{N} - \tilde{N} \|_{L_2(\mathcal{D})} \leq \frac{\epsilon}{5}$.

## A.3 Proof of Theorem 3.3

Let $\epsilon = \frac{1}{\text{poly}(d)}$, and let $N$ be a neural network of depth $k'$ such that $\|N - f\|_{L_2(\mu)} \leq \frac{\epsilon}{5}$. In the proof of Theorem 3.1 we constructed a network $\hat{N}$ of depth $3k' + 3$ such that $\|\hat{N} - f\|_{L_2(\mu)} \leq \epsilon$. The network $\hat{N}$ is such that in the first two layers the input $\mathbf{x}$ is transformed w.h.p. to the binary representation of $p(d)\tilde{\mathbf{x}}$. This transformation requires two layers, denoted by $\mathcal{N}_1$. Since the second layer in $\mathcal{N}_1$ does not have activation, it is combined with the next layer in $\hat{N}$. The next layers in $\hat{N}$, denoted by $\mathcal{N}_2$, implement a threshold circuit $T$ of depth $3k' + 1$ and width $\text{poly}(d)$. The depth of $\mathcal{N}_2$ is $3k' + 2$. Since the final layer of $\mathcal{N}_2$ does not have activation, it is combined with the next layer in $\hat{N}$. Finally, the output of $\hat{N}$ is obtained by computing a linear function over the outputs of $\mathcal{N}_2$.

Let $g : \{0,1\}^{d'} \to \{0,1\}$ be the function that $T$ computes. Note that $d' = \text{poly}(d)$. Assume that $g$ can be computed by a threshold circuit $T'$ of depth $k - 2$ and width $\text{poly}(d')$. By Lemma A.10, the threshold circuite $T'$ can be implemented by a neural network $\mathcal{N}_2'$ of depth $k - 1$ and width $\text{poly}(d')$. Consider the neural network $\hat{\mathcal{N}}$ obtained from $\hat{N}$ by replacing $\mathcal{N}_2$ with $\mathcal{N}_2'$. The depth of $\hat{\mathcal{N}}$ is $k$. The same arguments from the proof of Theorem 3.1 for showing that $\|\hat{N} - f\|_{L_2(\mu)} \leq \epsilon$ now apply on $\hat{\mathcal{N}}$, and hence $\|\hat{\mathcal{N}} - f\|_{L_2(\mu)} \leq \epsilon$. Therefore, $f$ can be approximated by a network of depth $k$, in contradiction to the assumption. Hence the function $g$ cannot be computed by a $\text{poly}(d')$-sized threshold circuit of depth $k - 2$.

## A.4 Proof of Theorem 3.4

Let $\epsilon = \frac{1}{\text{poly}(d)}$, and let $N$ be a neural network of depth $k'$ such that $\|N - f\|_{L_2(\mathcal{D})} \leq \frac{\epsilon}{5}$. In the proof of Theorem 3.2 we constructed a network $\hat{N}$ of depth $3k' + 3$ such that $\|\hat{N} - f\|_{L_2(\mathcal{D})} \leq \epsilon$. The structure of the network $\hat{N}$ is similar to the corresponding network from the proof of Theorem 3.1. Now, the proof follows the same lines as the proof of Theorem 3.3.

## A.5 Proof of Theorem 3.5

Let $f(\mathbf{x}) = g(\|\mathbf{x}\|)$ where $g : \mathbb{R} \to \mathbb{R}$. Let $\epsilon = \frac{1}{\text{poly}(d)}$. By Theorem 3.1, there is a neural network $N$ of a constant depth $k$, width $\text{poly}(d)$, and $\text{poly}(d)$-bounded weights, such that $\mathbb{E}_{\mathbf{x}\sim\mu}(N(\mathbf{x}) - f(\mathbf{x}))^2 \leq \left(\frac{\epsilon}{3}\right)^2$. Since $N$ has a constant depth, $\text{poly}(d)$ width and $\text{poly}(d)$-bounded weights, then it is $\text{poly}(d)$-Lipschitz. Also, as we show in the proof of Theorem 3.1, the network $N$ is bounded by some $B = \text{poly}(d)$, namely, for every $\mathbf{x} \in \mathbb{R}^d$ we have $|N(\mathbf{x})| \leq B$.

Let $r = \|\mathbf{x}\|$ and let $\mu_r$ be the distribution of $r$ where $\mathbf{x} \sim \mu$. Let $U(\mathbb{S}^{d-1})$ be the uniform distribution on the unit sphere in $\mathbb{R}^d$. Since $\mu$ is radial, we have

$$\left(\frac{\epsilon}{3}\right)^2 \geq \mathbb{E}_{\mathbf{x}\sim\mu}(N(\mathbf{x})-f(\mathbf{x}))^2 = \mathbb{E}_{\mathbf{z}\sim U(\mathbb{S}^{d-1})}\mathbb{E}_{r\sim\mu_r}(N(r\mathbf{z})-f(r\mathbf{z}))^2 = \mathbb{E}_{\mathbf{z}\sim U(\mathbb{S}^{d-1})}\mathbb{E}_{r\sim\mu_r}(N(r\mathbf{z})-g(r))^2.$$

Therefore, there is some $\mathbf{u} \in \mathbb{S}^{d-1}$ such that $\mathbb{E}_{r\sim\mu_r}(N(r\mathbf{u}) - g(r))^2 \leq \left(\frac{\epsilon}{3}\right)^2$. Let $N_\mathbf{u} : \mathbb{R} \to \mathbb{R}$ be such that $N_\mathbf{u}(t) = N(t\mathbf{u})$. It can be implemented by a network of depth $k$ that is obtained by preceding $N$ with a layer that computes $t \mapsto t\mathbf{u}$ (and does not have activation). Thus, $\mathbb{E}_{r\sim\mu_r}(N_\mathbf{u}(r) - g(r))^2 \leq \left(\frac{\epsilon}{3}\right)^2$. Let $h : \mathbb{R}^d \to \mathbb{R}$ be such that $h(\mathbf{x}) = N_\mathbf{u}(\|\mathbf{x}\|)$. Note that

$$\mathbb{E}_{\mathbf{x}\sim\mu}(h(\mathbf{x}) - f(\mathbf{x}))^2 = \mathbb{E}_{\mathbf{x}\sim\mu}(N_\mathbf{u}(\|\mathbf{x}\|) - g(\|\mathbf{x}\|))^2 = \mathbb{E}_{r\sim\mu_r}(N_\mathbf{u}(r) - g(r))^2 \leq \left(\frac{\epsilon}{3}\right)^2. \quad (11)$$

Since $\mu$ has an almost-bounded conditional density, then by Lemma A.4, there is $R_1 = \frac{1}{\text{poly}(d)}$ such that for every $i \in [d]$ we have

$$Pr_{\mathbf{x}\sim\mu}(x_i \in [-R_1, R_1]) \leq \frac{\epsilon^2}{72B^2}.$$

Hence,

$$Pr_{\mathbf{x}\sim\mu}(\|\mathbf{x}\| \leq R_1) \leq \frac{\epsilon^2}{72B^2}.$$

Also, since $\mu$ has an almost-bounded support, there exists $R_1 < R_2 = \text{poly}(d)$ such that

$$Pr_{\mathbf{x}\sim\mu}\left(\|\mathbf{x}\| \geq R_2\right) \leq \frac{\epsilon^2}{72B^2} \ .$$

Thus,

$$Pr_{r\sim\mu_r}(R_1 \leq r \leq R_2) \geq 1 - \frac{\epsilon^2}{36B^2} \ . \tag{12}$$

Since the network $N$ is bounded by $B$ then $N_{\mathbf{u}}$ is also bounded by $B$, namely, for every $t \in \mathbb{R}$ we have $|N_{\mathbf{u}}(t)| \leq B$. Moreover, since $N$ is $\text{poly}(d)$-Lipschitz, then $N_{\mathbf{u}}$ is also $\text{poly}(d)$-Lipschitz. Let $N'_{\mathbf{u}} : \mathbb{R} \to \mathbb{R}$ be such that

$$N'_{\mathbf{u}}(t) = \begin{cases} 0 & t \leq \frac{R_1}{2} \\ \frac{2N_{\mathbf{u}}(R_1)}{R_1} \cdot t - N_{\mathbf{u}}(R_1) & \frac{R_1}{2} < t \leq R_1 \\ N_{\mathbf{u}}(t) & R_1 < t \leq R_2 \\ -\frac{N_{\mathbf{u}}(R_2)}{R_2} \cdot t + 2N_{\mathbf{u}}(R_2) & R_2 < t \leq 2R_2 \\ 0 & t > 2R_2 \end{cases} \ .$$

Note that $N'_{\mathbf{u}}$ agrees with $N_{\mathbf{u}}$ on $[R_1, R_2]$, supported on $\left[\frac{R_1}{2}, 2R_2\right]$, bounded by $B$, and $\text{poly}(d)$-Lipschitz. Let $h' : \mathbb{R}^d \to \mathbb{R}$ be such that $h'(\mathbf{x}) = N'_{\mathbf{u}}(\|\mathbf{x}\|)$. We have

$$\mathbb{E}_{\mathbf{x}\sim\mu} \left(h'(\mathbf{x}) - h(\mathbf{x})\right)^2 = \mathbb{E}_{\mathbf{x}\sim\mu} \left(N'_{\mathbf{u}}(\|\mathbf{x}\|) - N_{\mathbf{u}}(\|\mathbf{x}\|)\right)^2 = \mathbb{E}_{r\sim\mu_r} \left(N'_{\mathbf{u}}(r) - N_{\mathbf{u}}(r)\right)^2 \ .$$

By Eq. 12 the functions $N_{\mathbf{u}}$ and $N'_{\mathbf{u}}$ agree w.p. at least $1 - \frac{\epsilon^2}{36B^2}$. Also, since both $N_{\mathbf{u}}$ and $N'_{\mathbf{u}}$ are bounded by $B$, we have $|N_{\mathbf{u}}(r) - N'_{\mathbf{u}}(r)| \leq 2B$ for every $r$. Hence, the above is at most

$$\frac{\epsilon^2}{36B^2} \cdot (2B)^2 + 0 = \left(\frac{\epsilon}{3}\right)^2 \ . \tag{13}$$

Now, we need the following Lemma.

**Lemma A.11.** *[7] Let $f : \mathbb{R} \to \mathbb{R}$ be a $\text{poly}(d)$-Lipschitz function supported on $[r,R]$, where $r = \frac{1}{\text{poly}(d)}$ and $R = \text{poly}(d)$. Then, for every $\delta = \frac{1}{\text{poly}(d)}$, there exists a neural network $\mathcal{N}$ of depth 3, width $\text{poly}(d)$, and $\text{poly}(d)$-bounded weights, such that*

$$\sup_{\mathbf{x}\in\mathbb{R}^d} |\mathcal{N}(\mathbf{x}) - f(\|\mathbf{x}\|)| \leq \delta \ .$$

Since $N'_{\mathbf{u}}$ is $\text{poly}(d)$-Lipschitz and supported on $\left[\frac{R_1}{2}, 2R_2\right]$, then by Lemma A.11 there exists a network $\mathcal{N}$ of depth 3, width $\text{poly}(d)$, and $\text{poly}(d)$-bounded weights , such that

$$\sup_{\mathbf{x}\in\mathbb{R}^d} |\mathcal{N}(\mathbf{x}) - N'_{\mathbf{u}}(\|\mathbf{x}\|)| \leq \frac{\epsilon}{3} \ .$$

Therefore, we have

$$\|\mathcal{N} - h'\|_{L_2(\mu)} \leq \|\mathcal{N} - h'\|_\infty = \sup_{\mathbf{x}\in\mathbb{R}^d} |\mathcal{N}(\mathbf{x}) - N'_{\mathbf{u}}(\|\mathbf{x}\|)| \leq \frac{\epsilon}{3} \ .$$

Combining the above with Eq. 11 and 13, we have

$$\|\mathcal{N} - f\|_{L_2(\mu)} \leq \|\mathcal{N} - h'\|_{L_2(\mu)} + \|h' - h\|_{L_2(\mu)} + \|h - f\|_{L_2(\mu)} \leq \frac{\epsilon}{3} + \frac{\epsilon}{3} + \frac{\epsilon}{3} = \epsilon \ .$$

### A.6 Proof of Theorem 3.6

**Lemma A.12.** *Let $f : \mathbb{R} \to \mathbb{R}$ be a function that can be implemented by a neural network of width $n$ and constant depth. Then, $f$ can be implemented by a network of width $\text{poly}(n)$ and depth 2.*

*Proof.* A neural network with input dimension 1, constant depth, and width $n$, is piecewise linear with at most $\mathrm{poly}(n)$ pieces ([29]). Therefore, $f$ consists of $m = \mathrm{poly}(n)$ linear pieces.

Let $-\infty = a_0 < a_1 < \ldots < a_{m-1} < a_m = \infty$ be such that $f$ is linear in every interval $(a_i, a_{i+1})$. For every $i \in [m]$ Let $\alpha_i$ be the derivative of $f$ in the linear interval $(a_{i-1}, a_i)$. Now, we have

$$f(t) = f(a_1) - \alpha_1 [a_1 - t]_+ + \sum_{2 \leq i \leq m-1} (\alpha_i [t - a_{i-1}]_+ - \alpha_i [t - a_i]_+) + \alpha_m [t - a_{m-1}]_+ .$$

Note that $f$ can be implemented by a network of depth 2 and width $\mathrm{poly}(n)$. In order to avoid bias in the output neuron, we implement the additive constant term $f(a_1)$ by adding a hidden neuron with fan-in 0 and bias 1, and connecting it to the output neuron with weight $f(a_1)$. $\qquad \square$

Let $\epsilon = \frac{1}{\mathrm{poly}(d)}$. Let $N : \mathbb{R}^d \to \mathbb{R}$ be a neural network of a constant depth and $\mathrm{poly}(d)$ width, such that $\mathbb{E}_{\mathbf{x} \sim \mu} (N(\mathbf{x}) - f(\mathbf{x}))^2 \leq \left(\frac{\epsilon}{d}\right)^2$. For $\mathbf{z} \in \mathbb{R}^{d-1}$ and $t \in \mathbb{R}$ we denote $\mathbf{z}_{i,t} = (z_1, \ldots, z_{i-1}, t, z_i, \ldots, z_{d-1}) \in \mathbb{R}^d$. Since $\mu$ is such that the components are drawn independently, then for every $i \in [d]$ we have

$$\mathbb{E}_{\mathbf{x} \sim \mu} (N(\mathbf{x}) - f(\mathbf{x}))^2 = \mathbb{E}_{\mathbf{z} \sim \mu_{[d] \setminus i}} \mathbb{E}_{t \sim \mu_i} (N(\mathbf{z}_{i,t}) - f(\mathbf{z}_{i,t}))^2 \leq \left(\frac{\epsilon}{d}\right)^2 ,$$

and therefore for every $i$ there exists $\mathbf{y} \in \mathbb{R}^{d-1}$ such that

$$\mathbb{E}_{t \sim \mu_i} (N(\mathbf{y}_{i,t}) - f(\mathbf{y}_{i,t}))^2 \leq \left(\frac{\epsilon}{d}\right)^2 .$$

Let $f_i' : \mathbb{R} \to \mathbb{R}$ such that

$$f_i'(t) = N(\mathbf{y}_{i,t}) - \sum_{j \in [d] \setminus \{i\}} f_j((\mathbf{y}_{i,t})_j) .$$

Note that

$$\mathbb{E}_{t \sim \mu_i} (f_i'(t) - f_i(t))^2 = \mathbb{E}_{t \sim \mu_i} \left( N(\mathbf{y}_{i,t}) - \sum_{j \in [d] \setminus \{i\}} f_j((\mathbf{y}_{i,t})_j) - f_i(t) \right)^2$$

$$= \mathbb{E}_{t \sim \mu_i} (N(\mathbf{y}_{i,t}) - f(\mathbf{y}_{i,t}))^2 \leq \left(\frac{\epsilon}{d}\right)^2 . \tag{14}$$

Now, the function $f_i'$ can be implemented by a neural network of depth 2 and width $\mathrm{poly}(d)$ as follows. First, note the by Lemma A.12 it is sufficient to show that $f_i'$ can be implemented by a network $N_i'$ of a constant depth and $\mathrm{poly}(d)$ width. Since $N$ is a network of constant depth, $\mathbf{y}$ is a constant, and $f_j((\mathbf{y}_{i,t})_j)$ for $j \in [d] \setminus \{i\}$ are also constants, implementing such $N_i'$ is straightforward.

Let $N'$ be the depth-2, width-$\mathrm{poly}(d)$ network such that $N'(\mathbf{x}) = \sum_{i \in [d]} f_i'(x_i)$. This network is obtained from the networks for $f_i'$. For every $i \in [d]$ let $g_i : \mathbb{R}^d \to \mathbb{R}$ be such that $g_i(\mathbf{x}) = f_i(x_i)$. Also, let $g_i' : \mathbb{R}^d \to \mathbb{R}$ be such that $g_i'(\mathbf{x}) = f_i'(x_i)$. Note that $f(\mathbf{x}) = \sum_{i \in [d]} g_i(\mathbf{x})$ and $N'(\mathbf{x}) = \sum_{i \in [d]} g_i'(\mathbf{x})$. Now, by Eq. 14, for every $i \in [d]$ we have

$$\mathbb{E}_{\mathbf{x} \sim \mu} (g_i'(\mathbf{x}) - g_i(\mathbf{x}))^2 = \mathbb{E}_{t \sim \mu_i} (f_i'(t) - f_i(t))^2 \leq \left(\frac{\epsilon}{d}\right)^2 .$$

Therefore, $\|g_i' - g_i\|_{L_2(\mu)} \leq \frac{\epsilon}{d}$.

Hence, we have

$$\|N' - f\|_{L_2(\mu)} = \left\| \sum_{i \in [d]} g_i' - \sum_{i \in [d]} g_i \right\|_{L_2(\mu)} \leq \sum_{i \in [d]} \|g_i' - g_i\|_{L_2(\mu)} \leq d \cdot \frac{\epsilon}{d} = \epsilon .$$

# B  Almost-bounded conditional density

In this section we show for some common distributions that they indeed have almost-bounded conditional densities.

## B.1  Gaussians, mixtures of Gaussians and Gaussian smoothing

We use the following property of conditional normal distributions.

**Lemma B.1.** *(e.g., [4]) Let $\mathcal{N}(\mu, \Sigma)$ be a multivariate normal distribution on $\mathbb{R}^d$. For $\mathbf{x} \in \mathbb{R}^d$ we partition $\mathbf{x}$ such that $\mathbf{x} = (x_a, x_b)$, where $x_a \in \mathbb{R}^q$ and $x_b \in \mathbb{R}^{d-q}$. Accordingly, we also partition $\mu = (\mu_a, \mu_b)$ and*

$$\Sigma = \begin{bmatrix} \Sigma_{aa} & \Sigma_{ab} \\ \Sigma_{ba} & \Sigma_{bb} \end{bmatrix} ,$$

*where the dimensions of the mean vectors and the covariance matrix sub-blocks are chosen to match the sizes of $x_a, x_b$. Let $\Lambda = \Sigma^{-1}$. We denote its partition that correspond to the partition of $\mathbf{x}$ by*

$$\Lambda = \begin{bmatrix} \Lambda_{aa} & \Lambda_{ab} \\ \Lambda_{ba} & \Lambda_{bb} \end{bmatrix} .$$

*Then, the distribution of $x_a$ conditional on $x_b = \mathbf{c}$ is the normal distribution $\mathcal{N}(\bar{\mu}, \bar{\Sigma})$, where*

$$\bar{\mu} = \mu_a - \Lambda_{aa}^{-1}\Lambda_{ab}(\mathbf{c} - \mu_b) = \mu_a + \Sigma_{ab}\Sigma_{bb}^{-1}(\mathbf{c} - \mu_b) ,$$

*and*

$$\bar{\Sigma} = \Lambda_{aa}^{-1} = \Sigma_{aa} - \Sigma_{ab}\Sigma_{bb}^{-1}\Sigma_{ba} .$$

**Proposition B.1.** *Let $\delta = \frac{1}{\text{poly}(d)}$. Let $\Sigma$ be a positive definite matrix of size $d \times d$ whose minimal eigenvalue is at least $\delta$. Let $\mu \in \mathbb{R}^d$. Then, the multivariate normal distribution $\mathcal{N}(\mu, \Sigma)$ has an almost-bounded conditional density.*

*Proof.* Let $\Lambda = \Sigma^{-1}$. Let $\lambda_1, \ldots, \lambda_d$ be the eigenvalues of $\Sigma$. The eigenvalues of $\Lambda$ are $\lambda_1^{-1}, \ldots, \lambda_d^{-1}$ and are at most $M = \frac{1}{\delta}$. Thus, $\text{trace}(\Lambda) = \sum_{i \in [d]} \lambda_i^{-1} \leq dM$. Since $\Lambda$ is positive definite then all entries on its diagonal are positive, and since their sum is bounded by $dM$, then we have $0 < \Lambda_{ii} \leq dM$ for every $i \in [d]$.

Let $\mathbf{x} \sim \mathcal{N}(\mu, \Sigma)$, let $\mathbf{c} \in \mathbb{R}^{d-1}$, and let $i \in [d]$. We now consider the conditional distribution $x_i \mid x_1, \ldots, x_{i-1}, x_{i+1}, \ldots, x_d = \mathbf{c}$. This conditional distribution corresponds to Lemma B.1 with $q = 1$. Namely, this is a univariate normal distribution with variance $\Lambda_{aa}^{-1}$ where $\Lambda_{aa} \in \mathbb{R}$. Since all entries on the diagonal of $\Lambda$ are bounded by $dM$, then the variance $\sigma^2$ of the conditional distribution satisfies $\sigma^2 \geq (dM)^{-1}$. Since the density of a univariate normal distribution with variance $\sigma^2$ is bounded by $\frac{1}{\sqrt{2\pi\sigma^2}}$, then the density of the conditional distribution is at most $\frac{1}{\sqrt{2\pi\sigma^2}} \leq \sqrt{\frac{dM}{2\pi}} = \text{poly}(d)$. $\qquad\square$

We now consider Gaussian mixtures.

**Proposition B.2.** *Let $\Sigma_1, \ldots, \Sigma_k$ be positive definite matrices with eigenvalues at least $\delta = \frac{1}{\text{poly}(d)}$. Let $\mu_1, \ldots, \mu_k$ be vectors in $\mathbb{R}^d$. For $j \in [k]$ let $f^j$ be the density function of the normal distribution $\mathcal{N}(\mu_j, \Sigma_j)$. Let $f$ be a density function such that $f(\mathbf{x}) = \sum_{j \in [k]} w_j f^j(\mathbf{x})$ with $\sum_{j \in [k]} w_j = 1$. Then, $f$ has an almost-bounded conditional density.*

*Proof.* Let $i \in [d]$ and let $\mathbf{c} \in \mathbb{R}^{d-1}$. For $t \in \mathbb{R}$ we denote $\mathbf{c}_{i,t} = (c_1, \ldots, c_{i-1}, t, c_i, \ldots, c_{d-1}) \in \mathbb{R}^d$. As we showed in the proof of Proposition B.1, there is $M = \text{poly}(d)$ (that depends on $\delta$) such that for every $j \in [k]$ we have

$$f^j_{i \mid [d]\setminus i}(t \mid \mathbf{c}) = \frac{f^j(\mathbf{c}_{i,t})}{\int_{\mathbb{R}} f^j(\mathbf{c}_{i,t})dt} \leq M .$$

Hence, we have

$$f_{i \mid [d]\setminus i}(t \mid \mathbf{c}) = \frac{\sum_{j \in [k]} w_j f^j(\mathbf{c}_{i,t})}{\int_{\mathbb{R}} \sum_{j \in [k]} w_j f^j(\mathbf{c}_{i,t})dt} \leq \frac{\sum_{j \in [k]} w_j M \int_{\mathbb{R}} f^j(\mathbf{c}_{i,t})dt}{\sum_{j \in [k]} w_j \int_{\mathbb{R}} f^j(\mathbf{c}_{i,t})dt} = M .$$

$\qquad\square$

Likewise, we show that the density obtained by applying Gaussian smoothing to a density function, has an almost-bounded conditional density.

**Proposition B.3.** *Let $\delta = \frac{1}{\text{poly}(d)}$, and let $\Sigma$ be a positive definite matrix of size $d \times d$ whose minimal eigenvalue is at least $\delta$. Let $g$ be the density function of the multivariate normal distribution $\mathcal{N}(\mathbf{0}, \Sigma)$. Let $f$ be a density function and let $f' = f \star g$ be the convolution of $f$ and $g$. That is, $f'$ is the density function obtained from $f$ by Gaussian smoothing. Then, $f'$ has an almost-bounded conditional density.*

*Proof.* Let $i \in [d]$ and let $\mathbf{c} \in \mathbb{R}^{d-1}$. For $t \in \mathbb{R}$ we denote $\mathbf{c}_{i,t} = (c_1, \ldots, c_{i-1}, t, c_i, \ldots, c_{d-1}) \in \mathbb{R}^d$. For $\mathbf{y} \in \mathbb{R}^d$, let $g^{\mathbf{y}} : \mathbb{R}^d \to \mathbb{R}$ be such that $g^{\mathbf{y}}(\mathbf{x}) = g(\mathbf{x} - \mathbf{y})$. Note that $g^{\mathbf{y}}$ is the density of the normal distribution $\mathcal{N}(\mathbf{y}, \Sigma)$. By the proof of Proposition B.1, there is $M = \text{poly}(d)$ (that depends on $\delta$) such that for every $\mathbf{y}$, and every $\mathbf{c}, t$ and $i$, we have

$$\frac{g(\mathbf{c}_{i,t} - \mathbf{y})}{\int_{\mathbb{R}} g(\mathbf{c}_{i,t} - \mathbf{y}) dt} = \frac{g^{\mathbf{y}}(\mathbf{c}_{i,t})}{\int_{\mathbb{R}} g^{\mathbf{y}}(\mathbf{c}_{i,t}) dt} = g^{\mathbf{y}}_{i|[d]\backslash i}(t|\mathbf{c}) \leq M \ . \tag{15}$$

Recall that

$$f'(\mathbf{c}_{i,t}) = (f \star g)(\mathbf{c}_{i,t}) = \int_{\mathbb{R}^d} f(\mathbf{y}) g(\mathbf{c}_{i,t} - \mathbf{y}) d\mathbf{y} \ .$$

Now, we have

$$
\begin{aligned}
f'_{i|[d]\backslash i}(t|\mathbf{c}) &= \frac{f'(\mathbf{c}_{i,t})}{\int_{\mathbb{R}} f'(\mathbf{c}_{i,t}) dt} = \frac{\int_{\mathbb{R}^d} f(\mathbf{y}) g(\mathbf{c}_{i,t} - \mathbf{y}) d\mathbf{y}}{\int_{\mathbb{R}} \left[ \int_{\mathbb{R}^d} f(\mathbf{y}) g(\mathbf{c}_{i,t} - \mathbf{y}) d\mathbf{y} \right] dt} \\
&\overset{(Eq.\ 15)}{\leq} \frac{\int_{\mathbb{R}^d} f(\mathbf{y}) M \left[ \int_{\mathbb{R}} g(\mathbf{c}_{i,t} - \mathbf{y}) dt \right] d\mathbf{y}}{\int_{\mathbb{R}} \left[ \int_{\mathbb{R}^d} f(\mathbf{y}) g(\mathbf{c}_{i,t} - \mathbf{y}) d\mathbf{y} \right] dt} \\
&= \frac{M \int_{\mathbb{R}^d} \int_{\mathbb{R}} f(\mathbf{y}) g(\mathbf{c}_{i,t} - \mathbf{y}) dt d\mathbf{y}}{\int_{\mathbb{R}^d} \int_{\mathbb{R}} f(\mathbf{y}) g(\mathbf{c}_{i,t} - \mathbf{y}) dt d\mathbf{y}} = M \ .
\end{aligned}
$$

$\square$

## B.2 Uniform distribution on the ball

In the cases of Gaussians, Gaussian mixtures, and Gaussian smoothing, we showed that the conditional density of $x_i | x_1, \ldots, x_{i-1}, x_{i+1}, \ldots, x_d = \mathbf{c}$ is bounded for every $\mathbf{c} \in \mathbb{R}^{d-1}$. Note that the definition of almost-bounded conditional density allows the conditional density to be greater than $M$ for some set of $\mathbf{c} \in \mathbb{R}^{d-1}$ with a small marginal probability. In the case of the uniform distribution over a ball in $\mathbb{R}^d$, we show that we cannot bound the conditional density for all $\mathbf{c} \in \mathbb{R}^{d-1}$, but we can bound it for a set in $\mathbb{R}^{d-1}$ with large marginal probability, which is sufficient by the definition of almost-bounded conditional density.

Let $\mu$ be the uniform distribution over the ball of a constant radius $R$ in $\mathbb{R}^d$. Let $\mathbf{c} \in \mathbb{R}^{d-1}$ be such that $\sum_{j \in [d-1]} c_j^2 = R^2 - \frac{1}{2^d}$. Let $i \in [d]$. For $t \in \mathbb{R}$, let $\mathbf{c}_{i,t} = (c_1, \ldots, c_{i-1}, t, c_i, \ldots, c_{d-1}) \in \mathbb{R}^d$. Note that $\mu_{i|[d]\backslash i}(t|\mathbf{c}) = 0$ for every $t$ such that $\sum_{j \in [d-1]} c_j^2 + t^2 > R^2$, namely, for every

$$|t| > \sqrt{R^2 - \sum_{j \in [d-1]} c_j^2} = \sqrt{\frac{1}{2^d}} = \frac{1}{2^{d/2}} \ .$$

Hence, the conditional density $\mu_{i|[d]\backslash i}(t|\mathbf{c}) = \frac{\mu(\mathbf{c}_{i,t})}{\mu_{[d]\backslash i}(\mathbf{c})}$ is uniform on the interval $\left[ -\frac{1}{2^{d/2}}, \frac{1}{2^{d/2}} \right]$. Therefore, we have

$$\mu_{i|[d]\backslash i}(t|\mathbf{c}) = \frac{1}{2 \cdot \frac{1}{2^{d/2}}} = 2^{\frac{d}{2}-1} \ .$$

Thus, for such $\mathbf{c}$ we cannot bound $\mu_{i|[d]\backslash i}(t|\mathbf{c})$ with a polynomial. However, as we show in the following proposition, the marginal probability to obtain such $\mathbf{c}$ is small, and $\mu$ has an almost-bounded conditional density.

**Proposition B.4.** *Let $\mu$ be the uniform distribution over the ball of radius $R \geq \frac{1}{\text{poly}(d)}$ in $\mathbb{R}^d$. Then, $\mu$ has an almost-bounded conditional density.*

*Proof.* Let $\epsilon = \frac{1}{\text{poly}(d)}$ and let $M = \frac{\sqrt{d}}{2R\sqrt{2\epsilon}}$. Let $i \in [d]$ and let $\mathbf{c} \in \mathbb{R}^{d-1}$. We denote $r = \sqrt{\sum_{j \in [d-1]} c_j^2}$. Note that $\mu_{i|[d]\setminus i}(t|\mathbf{c})$ is the uniform distribution over the interval $\left[-\sqrt{R^2 - r^2}, \sqrt{R^2 - r^2}\right]$. Hence, for every $t$ in this interval we have

$$\mu_{i|[d]\setminus i}(t|\mathbf{c}) = \frac{1}{2\sqrt{R^2 - r^2}} .$$

Note that if $r^2 \le R^2 - \frac{1}{4M^2}$ then $\mu_{i|[d]\setminus i}(t|\mathbf{c}) \le M$. Therefore, we have

$$Pr_{\mathbf{c} \sim \mu_{[d]\setminus i}} \left(\exists t \text{ s.t. } \mu_{i|[d]\setminus i}(t|\mathbf{c}) > M\right) \le Pr_{\mathbf{c} \sim \mu_{[d]\setminus i}} \left(\sum_{j \in [d-1]} c_j^2 > R^2 - \frac{1}{4M^2}\right)$$

$$\le Pr_{\mathbf{x} \sim \mu} \left(\sum_{j \in [d]} x_j^2 > R^2 - \frac{1}{4M^2}\right) .$$

Let $V_d(R)$ be the volume of the ball of radius $R$ in $\mathbb{R}^d$. Recall that $V_d(R) = V_d(1) \cdot R^d$. Note that the above equals to

$$\frac{1}{V_d(R)} \cdot \left(V_d(R) - V_d\left(\sqrt{R^2 - \frac{1}{4M^2}}\right)\right) = 1 - \frac{\left(\sqrt{R^2 - \frac{1}{4M^2}}\right)^d}{R^d}$$

$$= 1 - \left(1 - \frac{1}{4M^2R^2}\right)^{d/2} .$$

By Bernoulli's inequality, for every $z \ge -1$ and $y \ge 1$ we have $(1+z)^y \ge 1 + yz$. Therefore, the above is at most

$$1 - \left(1 - \frac{d}{8M^2R^2}\right) = \frac{d}{8M^2R^2} .$$

Plugging in $M = \frac{\sqrt{d}}{2R\sqrt{2\epsilon}}$, we obtain

$$Pr_{\mathbf{c} \sim \mu_{[d]\setminus i}} \left(\exists t \text{ s.t. } \mu_{i|[d]\setminus i}(t|\mathbf{c}) > M\right) \le \epsilon .$$

$\square$

### B.3 Distributions from existing depth-separation results

As we described in Section 1, the depth-separation result of [30], and the results that rely on it (e.g., [24, 32, 17]), are with respect to the uniform distribution on $[0,1]^d$. Thus, each component is chosen i.i.d. from the uniform distribution on the interval $[0,1]$, and therefore its conditional density is bounded by the constant 1.

The depth-separation result of [6] is for the function $f(\mathbf{x}_1, \mathbf{x}_2) = \sin(\pi d^3 \langle \mathbf{x}_1, \mathbf{x}_2 \rangle)$ with respect to the uniform distribution on $\mathbb{S}^{d-1} \times \mathbb{S}^{d-1}$, namely, both $\mathbf{x}_1$ and $\mathbf{x}_2$ are on the unit sphere. In [25], it is shown that this result can be easily reduced to a depth-separation result for the function $f(\mathbf{x}) = \sin(\frac{1}{2}\pi d^3 \|\mathbf{x}\|)$ and an $L_\infty$-type approximation. Moreover, from their proof it follows that this reduction applies also to an $L_2$ approximation with respect to an input $\mathbf{x} = \frac{\mathbf{x}_1 + \mathbf{x}_2}{2}$ where $\mathbf{x}_1$ and $\mathbf{x}_2$ are drawn i.i.d. from the uniform distribution on $\mathbb{S}^{d-1}$. We now show that this distribution has an almost-bounded conditional density. We first find the density function of $\|\mathbf{x}\|$.

**Lemma B.2.** *Let* $\mathbf{x} = \frac{\mathbf{x}_1 + \mathbf{x}_2}{2}$ *where* $\mathbf{x}_1$ *and* $\mathbf{x}_2$ *are drawn i.i.d. from the uniform distribution on* $\mathbb{S}^{d-1}$. *Then, the distribution of* $\|\mathbf{x}\|$ *has the density*

$$f_r(r) = \frac{1}{B\left(\frac{1}{2}, \frac{d-1}{2}\right)} 2^{d-1} r^{d-2} (1 - r^2)^{\frac{d-3}{2}} ,$$

*where* $B(\alpha, \beta) = \frac{\Gamma(\alpha)\Gamma(\beta)}{\Gamma(\alpha+\beta)}$ *is the beta function, and* $r \in (0,1)$.

*Proof.* Let $\mathbf{x} = \frac{\mathbf{x}_1 + \mathbf{x}_2}{2}$ where $\mathbf{x}_1$ and $\mathbf{x}_2$ are drawn i.i.d. from the uniform distribution on $\mathbb{S}^{d-1}$. Note that

$$\|\mathbf{x}\|^2 = \frac{1}{4}\left(\|\mathbf{x}_1\|^2 + \|\mathbf{x}_2\|^2 + 2\mathbf{x}_1^\top\mathbf{x}_2\right) = \frac{1}{4}\left(2 + 2\mathbf{x}_1^\top\mathbf{x}_2\right) = \frac{1}{2}\left(1 + \mathbf{x}_1^\top\mathbf{x}_2\right) . \qquad (16)$$

Since $\mathbf{x}_1$ and $\mathbf{x}_2$ are independent and uniformly distributed on the sphere, then the distribution of $\mathbf{x}_1^\top\mathbf{x}_2$ equals to the distribution of $(1, 0, \ldots, 0)\mathbf{x}_2$, which equals to the marginal distribution of the first component of $\mathbf{x}_2$. Let $z$ be the first component of $\mathbf{x}_2$. By standard results (cf. [8]), the distribution of $z^2$ is $\mathrm{Beta}(\frac{1}{2}, \frac{d-1}{2})$, namely, a Beta distribution with parameters $\frac{1}{2}, \frac{d-1}{2}$. Thus, the density of $z^2$ is

$$f_{z^2}(y) = \frac{1}{B\left(\frac{1}{2}, \frac{d-1}{2}\right)} y^{-\frac{1}{2}}(1-y)^{\frac{d-3}{2}} ,$$

where $B(\alpha, \beta) = \frac{\Gamma(\alpha)\Gamma(\beta)}{\Gamma(\alpha+\beta)}$ is the beta function, and $y \in (0, 1)$.

Performing a variable change, we obtain the density of $|z|$, which equals to the density of $|\mathbf{x}_1^\top\mathbf{x}_2|$.

$$f_{|\mathbf{x}_1^\top\mathbf{x}_2|}(y) = f_{|z|}(y) = f_{z^2}(y^2) \cdot 2y = \frac{1}{B\left(\frac{1}{2}, \frac{d-1}{2}\right)} y^{-1}(1-y^2)^{\frac{d-3}{2}} \cdot 2y = \frac{2}{B\left(\frac{1}{2}, \frac{d-1}{2}\right)}(1-y^2)^{\frac{d-3}{2}} ,$$

where $y \in (0, 1)$. Let $f_{\mathbf{x}_1^\top\mathbf{x}_2}$ be the density of $\mathbf{x}_1^\top\mathbf{x}_2$. Note that for every $y \in (-1, 1)$ we have $f_{\mathbf{x}_1^\top\mathbf{x}_2}(y) = f_{\mathbf{x}_1^\top\mathbf{x}_2}(-y)$. Hence, for every $y \in (-1, 1)$,

$$f_{\mathbf{x}_1^\top\mathbf{x}_2}(y) = \frac{1}{2}f_{|\mathbf{x}_1^\top\mathbf{x}_2|}(|y|) = \frac{1}{B\left(\frac{1}{2}, \frac{d-1}{2}\right)}(1-y^2)^{\frac{d-3}{2}} .$$

Performing a variable change again, we obtain the density of $\frac{1}{\sqrt{2}} \cdot \sqrt{1 + \mathbf{x}_1^\top\mathbf{x}_2}$.

$$f_{\frac{1}{\sqrt{2}} \cdot \sqrt{1+\mathbf{x}_1^\top\mathbf{x}_2}}(y) = f_{\mathbf{x}_1^\top\mathbf{x}_2}(2y^2 - 1) \cdot 4y = \frac{1}{B\left(\frac{1}{2}, \frac{d-1}{2}\right)}(1 - (4y^4 - 4y^2 + 1))^{\frac{d-3}{2}} \cdot 4y$$

$$= \frac{1}{B\left(\frac{1}{2}, \frac{d-1}{2}\right)}(2y)^{d-3}(1-y^2)^{\frac{d-3}{2}} \cdot 4y = \frac{1}{B\left(\frac{1}{2}, \frac{d-1}{2}\right)} 2^{d-1}y^{d-2}(1-y^2)^{\frac{d-3}{2}} .$$

Note that by Eq. 16 we have

$$\|\mathbf{x}\| = \sqrt{\frac{1 + \mathbf{x}_1^\top\mathbf{x}_2}{2}} ,$$

and therefore the density of $\|\mathbf{x}\|$ is

$$f_r(r) = f_{\frac{1}{\sqrt{2}} \cdot \sqrt{1+\mathbf{x}_1^\top\mathbf{x}_2}}(r) = \frac{1}{B\left(\frac{1}{2}, \frac{d-1}{2}\right)} 2^{d-1}r^{d-2}(1-r^2)^{\frac{d-3}{2}} .$$

$\square$

**Proposition B.5.** *Let* $\mathbf{x} = \frac{\mathbf{x}_1 + \mathbf{x}_2}{2}$ *where* $\mathbf{x}_1$ *and* $\mathbf{x}_2$ *are drawn i.i.d. from the uniform distribution on* $\mathbb{S}^{d-1}$. *Then the distribution of* $\mathbf{x}$ *has an almost-bounded conditional density.*

*Proof.* Let $\epsilon = \frac{1}{\mathrm{poly}(d)}$. Let $f_r$ be the distribution of $\|\mathbf{x}\|$. By Lemma B.2, we have

$$f_r(r) = \frac{1}{B\left(\frac{1}{2}, \frac{d-1}{2}\right)} 2^{d-1}r^{d-2}(1-r^2)^{\frac{d-3}{2}} . \qquad (17)$$

Let $\mu : \mathbb{R}^d \to \mathbb{R}$ be the density function on $\mathbb{R}^d$ that is induced by $f_r$. That is, $\mathbf{x} \sim \mu$ has the same distribution as $r\mathbf{u}$ where $r \sim f_r$ and $\mathbf{u}$ is distributed uniformly on $\mathbb{S}^{d-1}$. Let $i \in [d]$. For simplicity, we always assume in this proof that $d \geq 5$ (note that the definition of almost-bounded conditional density is not sensitive to the behavior of the density for small values of $d$).

We will first find $\delta_1, \delta_2 \leq \frac{1}{\mathrm{poly}(d)}$ such that

$$Pr_{\mathbf{c} \sim \mu_{[d]\setminus i}} (\delta_1 \leq \|\mathbf{c}\| \leq 1 - \delta_2) \geq 1 - \epsilon . \qquad (18)$$

Then, we will show that there is $M = \mathrm{poly}(d)$ such that for every $\mathbf{c} \in \mathbb{R}^{d-1}$ with $\delta_1 \leq \|\mathbf{c}\| \leq 1 - \delta_2$ and every $t \in (-1, 1)$ we have

$$\mu_{i|[d]\setminus i}(t|\mathbf{c}) \leq M . \tag{19}$$

We start with $\delta_2$. Note that

$$B\left(\frac{1}{2}, \frac{d-1}{2}\right) = \frac{\Gamma(\frac{1}{2})\Gamma(\frac{d-1}{2})}{\Gamma(\frac{d}{2})} \geq \frac{\Gamma(\frac{1}{2})\Gamma(\frac{d}{2}-1)}{\Gamma(\frac{d}{2})} = \frac{\Gamma(\frac{1}{2})}{\frac{d}{2}-1} \geq \frac{2\Gamma(\frac{1}{2})}{d} = \frac{2\sqrt{\pi}}{d} \geq \frac{1}{d} . \tag{20}$$

Let $\delta_2 = 1 - \sqrt{1 - \frac{\epsilon}{32d}}$. By the above and Eq. 17, for every $r \in (1 - \delta_2, 1)$ we have

$$f_r(r) \leq d2^{d-1} r^{d-2}(1 - r^2)^{\frac{d-3}{2}} \leq d2^{d-1}\left(1 - (1-\delta_2)^2\right)^{\frac{d-3}{2}} = d2^{d-1}\left(\frac{\epsilon}{32d}\right)^{\frac{d-3}{2}} .$$

Hence,

$$Pr_{\mathbf{c} \sim \mu_{[d]\setminus i}}\left(\|\mathbf{c}\| \geq 1 - \delta_2\right) \leq Pr_{r \sim f_r}\left(r \geq 1 - \delta_2\right) \leq d \cdot 2^{d-1}\left(\frac{\epsilon}{32d}\right)^{\frac{d-3}{2}} \cdot \delta_2$$

$$\leq d \cdot 4^{\frac{d-3}{2}} \cdot 4\left(\frac{\epsilon}{32d}\right)^{\frac{d-3}{2}} = 4d\left(\frac{\epsilon}{8d}\right)^{\frac{d-3}{2}} \leq 4d \cdot \frac{\epsilon}{8d} = \frac{\epsilon}{2} . \tag{21}$$

We now turn to $\delta_1$. By [8], the marginal distribution $\mathbf{c} \sim \mu_{[d]\setminus i}$ is such that $\|\mathbf{c}\| = r\alpha$, where $r$ and $\alpha$ are independent, $r \sim f_r$, and $\alpha^2 \sim \mathrm{Beta}\left(\frac{d-1}{2}, \frac{1}{2}\right)$, namely, a Beta distribution with parameters $\frac{d-1}{2}, \frac{1}{2}$. Hence, we have

$$Pr_{\mathbf{c} \sim \mu_{[d]\setminus i}}\left(\|\mathbf{c}\| \leq \delta_1\right) \leq Pr_{r \sim f_r}\left(r \leq \sqrt{\delta_1}\right) + Pr_{\beta \sim \mathrm{Beta}\left(\frac{d-1}{2}, \frac{1}{2}\right)}\left(\sqrt{\beta} \leq \sqrt{\delta_1}\right) . \tag{22}$$

We now bound the two part of the above right hand side. For $\delta = \frac{\epsilon}{16d}$, we have by Eq. 17 and 20 that for every $r \in (0, \delta)$,

$$f_r(r) \leq d2^{d-1} r^{d-2}(1 - r^2)^{\frac{d-3}{2}} \leq d2^{d-1}\delta^{d-2} = d2^{d-1}\left(\frac{\epsilon}{16d}\right)^{d-2} = 2d\left(\frac{\epsilon}{8d}\right)^{d-2} \leq 2d \cdot \frac{\epsilon}{8d} = \frac{\epsilon}{4} .$$

Thus, for $\delta_1 = \delta^2$ we have

$$Pr_{r \sim f_r}\left(r \leq \sqrt{\delta_1}\right) = Pr_{r \sim f_r}\left(r \leq \delta\right) \leq \delta \cdot \frac{\epsilon}{4} \leq \frac{\epsilon}{4} . \tag{23}$$

Moreover, we have

$$Pr_{\beta \sim \mathrm{Beta}\left(\frac{d-1}{2}, \frac{1}{2}\right)}\left(\sqrt{\beta} \leq \sqrt{\delta_1}\right) = Pr_{\beta \sim \mathrm{Beta}\left(\frac{d-1}{2}, \frac{1}{2}\right)}\left(\beta \leq \delta_1\right)$$

$$= \int_0^{\delta_1} \frac{1}{B\left(\frac{d-1}{2}, \frac{1}{2}\right)} \beta^{\frac{d-1}{2}-1}(1-\beta)^{\frac{1}{2}-1} d\beta$$

$$\leq \delta_1 \cdot \frac{1}{B\left(\frac{d-1}{2}, \frac{1}{2}\right)} \cdot \delta_1^{\frac{d-3}{2}} \cdot \frac{1}{\sqrt{1-\delta_1}}$$

Since $0 < \delta_1 \leq \frac{1}{2}$, and by plugging in Eq. 20, the above is at most

$$d \cdot \delta_1^{\frac{d-1}{2}} \cdot \frac{1}{\sqrt{1 - \frac{1}{2}}} \leq d \cdot \delta_1^{\frac{1}{2}} \cdot \sqrt{2} = \sqrt{2}d \cdot \frac{\epsilon}{16d} \leq \frac{\epsilon}{4} . \tag{24}$$

Combining Eq. 22, 23 and 24, we have

$$Pr_{\mathbf{c} \sim \mu_{[d]\setminus i}}\left(\|\mathbf{c}\| \leq \delta_1\right) \leq \frac{\epsilon}{2} .$$

Then, Eq. 18 follows by combining the above with Eq. 21. Thus, it remains to show that there is $M = \mathrm{poly}(d)$ such that for every $\mathbf{c} \in \mathbb{R}^{d-1}$ with $\delta_1 \leq \|\mathbf{c}\| \leq 1 - \delta_2$ and every $t \in (-1, 1)$, Eq. 19 holds.

Let $A_d$ be the surface area of the unit sphere in $\mathbb{R}^d$. Note that for every $\mathbf{x} \neq \mathbf{0}$ in the unit ball, we have

$$\mu(\mathbf{x}) = \frac{f_r(\|\mathbf{x}\|)}{\|\mathbf{x}\|^{d-1} A_d} = \frac{1}{B\left(\frac{1}{2}, \frac{d-1}{2}\right)} 2^{d-1} \|\mathbf{x}\|^{d-2} (1 - \|\mathbf{x}\|^2)^{\frac{d-3}{2}} \cdot \frac{1}{\|\mathbf{x}\|^{d-1} A_d}$$

$$= \frac{1}{A_d B\left(\frac{1}{2}, \frac{d-1}{2}\right)} 2^{d-1} (1 - \|\mathbf{x}\|^2)^{\frac{d-3}{2}} \cdot \frac{1}{\|\mathbf{x}\|} .$$

For $t \in \mathbb{R}$, we denote $\mathbf{c}_{i,t} = (c_1, \ldots, c_{i-1}, t, c_i, \ldots, c_{d-1}) \in \mathbb{R}^d$. Now, we have

$$\mu_{[d] \setminus i}(\mathbf{c}) = \int_{-1}^{1} \mu(\mathbf{c}_{i,t}) dt = \int_{-\sqrt{1-\|\mathbf{c}\|^2}}^{\sqrt{1-\|\mathbf{c}\|^2}} \frac{1}{A_d B\left(\frac{1}{2}, \frac{d-1}{2}\right)} 2^{d-1} (1 - (\|\mathbf{c}\|^2 + t^2))^{\frac{d-3}{2}} \cdot \frac{1}{\sqrt{\|\mathbf{c}\|^2 + t^2}} dt .$$

Performing the variable change $z = \sqrt{\|\mathbf{c}\|^2 + t^2}$, the above equals

$$2 \int_{\|\mathbf{c}\|}^{1} \frac{1}{A_d B\left(\frac{1}{2}, \frac{d-1}{2}\right)} 2^{d-1} (1 - z^2)^{\frac{d-3}{2}} \cdot \frac{1}{z} \cdot \frac{z}{\sqrt{z^2 - \|\mathbf{c}\|^2}} dz$$

$$\geq 2 \int_{\|\mathbf{c}\|}^{1} \frac{1}{A_d B\left(\frac{1}{2}, \frac{d-1}{2}\right)} 2^{d-1} (1 - z^2)^{\frac{d-3}{2}} \cdot \frac{1}{z} dz$$

$$= \frac{2^d}{A_d B\left(\frac{1}{2}, \frac{d-1}{2}\right)} \int_{\|\mathbf{c}\|}^{1} (1 + z)^{\frac{d-3}{2}} (1 - z)^{\frac{d-3}{2}} \cdot \frac{1}{z} dz$$

$$\geq \frac{2^d}{A_d B\left(\frac{1}{2}, \frac{d-1}{2}\right)} (1 + \|\mathbf{c}\|)^{\frac{d-3}{2}} \int_{\|\mathbf{c}\|}^{1} (1 - z)^{\frac{d-3}{2}} dz .$$

By plugging in

$$\int_{\|\mathbf{c}\|}^{1} (1 - z)^{\frac{d-3}{2}} dz = -\frac{(1 - z)^{\frac{d-3}{2} + 1}}{\frac{d-3}{2} + 1} \Bigg|_{\|\mathbf{c}\|}^{1} = \frac{2(1 - \|\mathbf{c}\|)^{\frac{d-1}{2}}}{d - 1} ,$$

we get

$$\frac{2^{d+1} (1 + \|\mathbf{c}\|)^{\frac{d-3}{2}} (1 - \|\mathbf{c}\|)^{\frac{d-1}{2}}}{A_d B\left(\frac{1}{2}, \frac{d-1}{2}\right) (d - 1)} .$$

Hence,

$$\mu_{i | [d] \setminus i}(t | \mathbf{c}) = \frac{\mu(\mathbf{c}_{i,t})}{\mu_{[d] \setminus i}(\mathbf{c})}$$

$$\leq \frac{1}{A_d B\left(\frac{1}{2}, \frac{d-1}{2}\right)} 2^{d-1} (1 - \|\mathbf{c}_{i,t}\|^2)^{\frac{d-3}{2}} \cdot \frac{1}{\|\mathbf{c}_{i,t}\|} \cdot \frac{A_d B\left(\frac{1}{2}, \frac{d-1}{2}\right) (d - 1)}{2^{d+1} (1 + \|\mathbf{c}\|)^{\frac{d-3}{2}} (1 - \|\mathbf{c}\|)^{\frac{d-1}{2}}}$$

$$= (1 - \|\mathbf{c}_{i,t}\|^2)^{\frac{d-3}{2}} \cdot \frac{1}{\|\mathbf{c}_{i,t}\|} \cdot \frac{d - 1}{2^2 (1 + \|\mathbf{c}\|)^{\frac{d-3}{2}} (1 - \|\mathbf{c}\|)^{\frac{d-1}{2}}}$$

$$\leq (1 + \|\mathbf{c}\|)^{\frac{d-3}{2}} (1 - \|\mathbf{c}\|)^{\frac{d-3}{2}} \cdot \frac{1}{\|\mathbf{c}\|} \cdot \frac{d - 1}{4 (1 + \|\mathbf{c}\|)^{\frac{d-3}{2}} (1 - \|\mathbf{c}\|)^{\frac{d-1}{2}}}$$

$$= \frac{1}{\|\mathbf{c}\|} \cdot \frac{d - 1}{4(1 - \|\mathbf{c}\|)} .$$

Now, since $\delta_1 \leq \|\mathbf{c}\| \leq 1 - \delta_2$, the above is at most

$$\frac{1}{\delta_1} \cdot \frac{d - 1}{4 \delta_2} \leq \mathrm{poly}(d) .$$

$\square$

[7] showed separation between depth 2 and 3 for a $\mathrm{poly}(d)$-Lipschitz radial function $f : \mathbb{R}^d \to \mathbb{R}$ with respect to a distribution with density

$$\mu(\mathbf{x}) = \left(\frac{R_d}{\|\mathbf{x}\|}\right)^d J_{d/2}^2(2\pi R_d \|\mathbf{x}\|) \,,$$

where $R_d$ is the radius of the unit-volume Euclidean ball in $\mathbb{R}^d$, and $J_{d/2}$ is a Bessel function of the first kind. An analysis of its conditional density requires some investigation of Bessel functions and is not included here. However, it is not hard to show that for every polynomial $p(d)$, there is a distribution $\mu'$ (obtained by applying Gaussian smoothing to $\mu$ and has an almost-bounded conditional density by Proposition B.3), such that the function $f$ can be expressed by a depth-3 network but cannot be approximated by a depth-2 network with a Lipschitz constant bounded by $p(d)$. This follows from the fact that if there was a Lipschitz approximating network under $\mu'$, it would also be approximating under the slightly different distribution $\mu$.