[Reviews · NeurIPS 2020]

Review 1

Summary and Contributions: This paper considers parallels between depth separation for neural networks and the corresponding problem studied in circuit complexity. They main distinction is that for circuits the inputs and output are binary, while for NN they are real valued. Theorem 3.1 (and 3.2) show that under some regularity conditions on the distribution w.r.t. which approximation happens, one can assume that the weights are polynomially bounded with only moderate increase (factor 3) in the depth. The main idea of the argument is to reduce weights of magnitude 2^poly(d) to poly(d) by performing operations on the binary representation. There are various additional technical issues that are overcome in the argument. Theorems 3.3 and 3.4 show that depth separation for neural networks implies depth separation for threshold circuits over the hypercube. The paper proves the contrapositive, i.e. assumes there is no depth separation for threshold circuits of certain depths and then argues lack of depth separation for NN. The argument converts a NN of depth k' to a threshold circuit, and one then simulates the threshold circuit with smaller depth, which then converts back to a NN of smaller depth k < k'. Overall the paper makes some nice observations, and I think it should probably be accepted.

Strengths: Stronger connections between neural network approximation and the circuit complexity literature are badly needed, as it often feels that researchers are reinventing the wheel. This paper makes some such connections and will hopefully stimulate further work in this direction.

Weaknesses: The basic ideas underlying the results are not groundbreaking conceptually. Requiring only polynomial versus not polynomial in the separation definition is a bit limiting. One might hope for more fine-grained results.

Correctness: The mathematical arguments appear to be correct.

Clarity: The paper would definitely benefit from clearer mathematical exposition as well as better conceptual organization. For example: line 209: instead of putting \exists there, better to write \sup_t theorem 3.1: add in the w.r.t. to \mu when talking about the approximation section 4.2 would benefit from slightly more details / explanation the entirety of section 4 would benefit from stating a couple of lemmas on the conversion which are then reused as needed

Relation to Prior Work: The connection to prior work is reasonably comprehensive.

Reproducibility: Yes

Additional Feedback:


Review 2

Summary and Contributions: This paper studies the depth separation for the approximation ability of deep fully-connected ReLU networks. Different from the previous works, this work tries to consider the separation of neural networks of two general depths k, k'. The first result shows that if a function can be approximated by a network of depth k, width poly(d), it can also be approximated by a network of depth=3k+3, width=poly(d), and weights bounded by poly(d). This theorem suggests that the magnitude of weights is not extremely important for approximating functions with the output values bounded by poly(d). The second results show that if a function reaches the depth separation for networks of depths k and k'. Then there exists a boolean function g reaching the depth separation for threshold circuits of depths k-2, 3k'+1. The depth separation of threshold circuits has been well studied in the theoretical computer science community. Especially the concept of the nature-proof barrier is developed to explain the difficulty to obtain this separation. Hence, the second result provide us an understanding of the difficulty to obtain the depth separation for neural networks

Strengths: As far as I can tell, this work makes a significant contribution for us understanding the depth separation for neural networks. Basically, it provides an explanation of why it is so hard to obtain a depth separation for neural networks of general depths. The first theorem is also very interesting. It basically implies that to approximate any bounded functions, the network weights only require in the order of poly(d).

Weaknesses: The only weakness of this paper is that all the statements are too abstract. It is very hard to get some intuitions behind these results. For example, bounded functions can oscillate exponentially and has an exp(d) Lipschitz norm. However, the first theorem suggests they can be approximated with only poly(d) weights. This looks very counterintuitive. How should we understand it?

Correctness: Due to the very limited review time, I only take a quick look the appendix. The proof looks reasonable to me, but I am not sure the proofs are correct.

Clarity: The paper is well written, especially the connection with the previous works. I can easily follow authors' idea.

Relation to Prior Work: Yes!

Reproducibility: Yes

Additional Feedback: --- After rebuttal --- I read the authors' response, and my concern was addressed. I also suggest to add these explanations to the paper in the revised version, which is much helpful for readers to understand the theorem.


Review 3

Summary and Contributions: This paper proves two main results: 1. If one can prove a separation between feedforward ReLU networks of depth 4 and some larger depth, then it shows a separation between threshold circuits of depth 2 (with arbitrarily large wights) and some larger depth. 2. Under mild assumptions, if the function can be approximated by a network with ReLU activation, width poly(d), constant depth k, and arbitrarily large weights, then it can be approximated by a network with ReLU activation, width poly(d), depth 3k+3.

Strengths: This work connected the question of separation of the approximation power of neural networks to an old conjecture in circuit complexity.

Weaknesses: It is not clear what are the implications of these results to practical neural networks. This paper seems to imply that, there may not be a polynomial--exponential separation between networks of depth 4 and higher depth (assuming there is no separation of corresponding thresholding circuit). However, as the authors commented in this paper, there are ways to massage the architecture and feature distribution to circumvent the assumptions of these results. Moreover, even if there is no polynomial--exponential separation between networks of different depth, it does not imply going deeper has no approximation benefits since there could possibly be low-degree-polynomial--high-degree-polynomial separation. Note that in practical neural networks, it makes a huge difference if the width/weights scale linear in d instead of being constant.

Correctness: The reviewer does not have the expertise to assess the correctness of the proofs.

Clarity: Yes

Relation to Prior Work: Yes

Reproducibility: Yes

Additional Feedback: Personally, I don't see how the results are interesting or useful. But perhaps this result is interesting to circuit complexity community. This falls out of my expertise, so I don't hold very strong opinion on whether to accept or reject. Post rebuttal: My feeling is that, if we only look at its implications for neural networks, these results are not completely satisfying. Even if it is hard to show a polynomial-exponential separation between depth k and k' networks in the setting as the authors described, there may be ways to show separation in other settings (if the probability density assumption is violated) or other kinds of separations. However, as other reviewers pointed out, the reduction from neural networks to circuits (and introducing it to NeurIPS community) is an interesting and important contribution. I am convinced by this point and I will raise my score to 6.


Review 4

Summary and Contributions: This paper studies the question of proving depth separation for neural networks with real inputs and the ReLU activation function. It shows that constructing a real function that can be computed by a polysize NN of depth k' but not computed by a polysize network of depth k<k' where k and k' are appropriate constants will imply a depth separation between threshold circuits  (namely a Boolean function that can be efficiently be computed at depth r' but has not polysize threshold circuit of depth r<r) a notorious open problem in circuit complexity. A key observation is that polynomially bounded real function that can be efficiently computed by an NN of depth k can be efficiently computed in depth 3k with polynomially bounded weights.  Then they use the fact that integer arithmetic can be done with threshold circuits of bounded depth to "replace" the second to one before last layers with a TC T. This T cannot be compressed to a lower depth circuit of poly-size or we will get a contradiction to our assumption about the depth separation of the original NN.

Strengths: The paper considers two natural problem in the theoretical analysis of neural networks: the effect of depth on expressiveness and the role of large weights in implementing function with "small networks". Along the way if offers new connection to machinery in circuit complexity that could have further applications.

Weaknesses: The paper mostly uses known results without introducing new techniques.

Correctness: Claims and methods appear correct.

Clarity: The paper is well written. The authors do a great job in describing the problem, its history their results and their ideas.

Relation to Prior Work: (mostly) Yes: see above.

Reproducibility: Yes

Additional Feedback: 1) The authors keep repeating the natural proof barrier assumption in a rather cumbersome way. There is some debate among complexity theory about how serious of an obstacle this barrier actually is. Stating that showing a depth separation between NN will show a depth separation between TC is a strong enough evidence that the problem is likely to be hard. After rebuttal: my evaluation remains the same. 2) Do we know non-constructively that there exist real functions that cannot be expressed efficiently by a neural network at depth k but can at depth k+1? 3) The paper [17] assumes the depth is constant and that the fanout of each gate is one. This limitation should be stated. 4) The simulation of GHR [10] can also be extended to the case of arbitrary depth (that can depend on n and does not have to be a constant). See "Simulating Threshold Circuits by Majority Circuits" by Goldmann and Karpisnki. 5) The authors keep using the max width (=size of largest layer) in estimating the complexity of the neural network. Why not use the total size (total number of neurons) instead? 6) In the definition of threshold circuit requirement (2) is redundant. There is always a TC satisfying 1,3,4 with integer weights (and the same size): we can assume wlog all weights are integral. 7) I have found the notation of I in Theorem 3.1 somewhat cryptic. For example, why does p depend on d and not R? 8) The discussion between Remark 3.4 to Theorem 3.5 is largely redundant repeating previously stated observations. 9) Do the results of the authors hold for other approximation measures of real functions such asL_1 and L_{\infinity}? 10) I would consider removing the discussion between 92-102. It is somewhat convoluted. One gets a feeling as if the authors are trying to attenuate their results so to encourage people to keep working on the intersection of neural networks and circuit complexity: I personally did not find it convincing.  Perhaps instead the authors could identify questions related to the current paper that do not entail solving long standing open questions in Boolean circuit complexity.

[Author Response · NeurIPS 2020]

We thank the reviewers for their helpful comments and we will fix accordingly. Below we address some comments and questions that have been raised.

**Reviewers 1 and 3**

The reviewers wrote that requiring only polynomial versus not polynomial in the separation definition is a bit limiting. The benefits of depth can be studies from different angles. We focus on the notion of depth-separation that has been extensively studied in many prior works, and requires polynomial vs. exponential width. We agree that other notions of depth-separation should also be explored, and we mention in the "related work" section an example for such a different notion that has been studied in [23,16,31].

**Reviewer 2**

The reviewer asked: "bounded functions can oscillate exponentially and has an $\exp(d)$ Lipschitz norm. However, the first theorem suggests they can be approximated with only $\text{poly}(d)$ weights. This looks very counterintuiting. How should we understand it?". This observation is indeed surprising, however, it has a simple intuitive explanation. Consider a univariate function $f : \mathbb{R} \to \mathbb{R}$ that is expressible by a neural network $N$ of constant depth and $\text{poly}(d)$ width (for some integer $d$). Such a function is piecewise-linear with a $\text{poly}(d)$ number of pieces. Since the weights in $N$ might be exponential in $d$, then some of the linear pieces might have exponential derivatives. Thus, $f$ might oscillate quickly in some intervals. However, since $f$ is bounded, then an interval where $f$ has an exponential derivative must be very small (exponentially small). Hence, $f$ consists of $\text{poly}(d)$ linear pieces, but may oscillate quickly only in exponentially-small intervals. A network of constant depth and $\text{poly}(d)$ weights cannot approximate such $f$ in the $L_\infty$ sense. However, since we assume that the input distribution $\mu$ is not too concentrated in very small intervals, then it is possible to approximate $f$ in the $L_2(\mu)$ sense. The case where $f : \mathbb{R}^d \to \mathbb{R}$ is more complicated, but follows the same intuition.

**Reviewer 4**

Thanks for your feedback and comments, we will incorporate them into the final version. Below we address your specific questions:

(2) Not for $k > 2$.

(5) Since we focus on constant-depth networks, then it does not matter. We will comment on that in the final version.

(7) There is no special reason for this choice of notations. We will change.

(9) The results hold also for approximation w.r.t. $L_1(\mu)$ (it follows easily from our proof), but do not hold for approximation w.r.t. $L_\infty$.

[Meta-Review · NeurIPS 2020]

This paper makes connections between circuit complexity and expressiveness of neural networks as a function of their depth. Results in this paper relate the problem of "depth separation" (showing that adding depth increases the expressiveness) to circuit lower bounds for threshold circuits. They also prove that under some conditions on the distribution w.r.t. which the approx happens, one can assume that the weights of the network are polynomially bounded. The reviewers found the research direction of this paper compelling: making connections between neural networks and circuit complexity. I recommend this paper for acceptance.